# Structural mechanism of R2D2 and Loqs-PD synergistic modulation on *Dm*Dcr-2 oligomers

Ting Deng[1,4], Shichen Su[1,4], Xun Yuan[2], Jinqiu He[1], Ying Huang [2], Jinbiao Ma [1] ✉ & Jia Wang [3] ✉

Small interference RNAs are the key components of RNA interference, a conserved RNA silencing or viral defense mechanism in many eukaryotes. In *Drosophila melanogaster*, Dicer-2 (*Dm*Dcr-2)-mediated RNAi pathway plays important roles in defending against viral infections and protecting genome integrity. During the maturation of siRNAs, two cofactors can regulate *Dm*Dcr-2's functions: Loqs-PD that is required for dsRNA processing, and R2D2 that is essential for the subsequent loading of siRNAs into effector Ago2 to form RISC complexes. However, due to the lack of structural information, it is still unclear whether R2D2 and Loqs-PD affect the functions of *Dm*Dcr-2 simultaneously. Here we present several cryo-EM structures of *Dm*Dcr-2/R2D2/Loqs-PD complex bound to dsRNAs with various lengths by the Helicase domain. These structures revealed that R2D2 and Loqs-PD can bind to different regions of *Dm*Dcr-2 without interfering with each other. Furthermore, the cryo-EM results demonstrate that these complexes can form large oligomers and assemble into fibers. The formation and depolymerization of these oligomers are associated with ATP hydrolysis. These findings provide insights into the structural mechanism of *Dm*Dcr-2 and its cofactors during siRNA processing.

RNA interference (RNAi) is a conserved post-transcriptional regulatory mechanism mediated by small RNAs. It plays vital roles in various biological processes such as gene regulation, tumor progression, antiviral defense, and controlling transposable elements in the genome across eukaryotic organisms[1–4]. In this pathway, Argonaute (Ago) proteins, which are components of the RNA-induced silencing complex (RISC), identify target RNAs by complementary base pairing using its bound small noncoding RNAs. The small RNAs can be microRNAs (miRNAs) generated from short hairpin RNA (pre-miRNA) or small interfering RNAs (siRNAs) from double-stranded RNA (dsRNA), both of which are 21–25 nucleotide (nt) duplexes with phosphate at 5′ end and a 2-nt

overhang ended with hydroxyl termini at 3′ end[5]. In mammals, miRNA and siRNA are generated by the same RNase-III enzyme Dicer, while in *Drosophila melanogaster*, they are generally not only produced by different Dicer (*Dm*Dcr-1 and *Dm*Dcr-2) but also sorted to functionally distinct Argonaute proteins as *Dm*Ago1 and *Dm*Ago2, respectively[6].

Dicer, a member of the RNase-III family of enzymes, is highly conserved across eukaryotes. It consists of several domains, including an N-terminal DExD/H helicase (Helicase) domain, cap domains (Platform domain and PAZ domain), a core region with two RNase-III domains, and C-terminal double-stranded RNA-binding domains (dsRBD). The arrangement and collaboration of these domains are

[1]State Key Laboratory of Genetic Engineering, Collaborative Innovation Centre of Genetics and Development, Department of Biochemistry and Biophysics, Institute of Plant Biology, School of Life Sciences, Fudan University, 200438 Shanghai, China. [2]Shanghai Key Laboratory of Biliary Tract Disease Research, Shanghai Research Center of Biliary Tract Disease, Department of General Surgery, Xinhua Hospital, Affiliated with Shanghai Jiao Tong University School of Medicine, Shanghai, China. [3]Ministry of Education Key Laboratory of Protein Sciences, Beijing Advanced Innovation Center for Structural Biology, Beijing Frontier Research Center of Biological Structures, School of Life Sciences, Tsinghua University, Beijing, China. [4]These authors contributed equally: Ting Deng, Shichen Su. ✉e-mail: majb@fudan.edu.cn; wangjia2016@tsinghua.edu.cn

crucial in determining the precise length and structural characteristics of the RNA molecules cleaved by RNase III[7–11]. The helicase domain in *Hs*Dicer (Human Dicer) and *Dm*Dcr-1 exhibits a low affinity for dsRNA-binding and lacks the capability to hydrolyze ATP. Consequently, their primary substrate is limited to short hairpin-shaped pre-miRNA. In contrast, the Helicase domain of *Dm*Dcr-2 differs from *Hs*Dicer or *Dm*Dcr-1 as it can bind to dsRNA with high affinity and facilitates ATP hydrolysis to transport dsRNA substrates to the RNA processing center[12–14].

When the small RNA duplex precursors are processed by Dicer proteins, the cofactor dsRNA-binding proteins (dsRBPs) composed of multiple dsRBDs assist Dicer to complete the dicing process and transfer the products to Argonaute proteins[15]. Among these cofactors, TRBP/PACT in mammals and R2D2/Loquacious (Loqs) in *Drosophila* have been well-studied. These cofactors bind to Dicer at a stoichiometric ratio and affect its cleavage activity, product length, and product delivery etc.[16–19]. All dsRBDs adopt a common α–β–β–β–α-fold, but they can be divided into two distinct classes based on their dsRNA-binding ability. Type A dsRBDs share a similar dsRNA-binding way that is sequence-independent, whereas type B dsRBDs have no dsRNA-binding activity[20,21]. TRBP/PACT and Loqs-PA/Loqs-PB have two type A dsRBDs and one type B dsRBD that bind *Hs*Dicer and *Dm*Dcr-1, respectively[22–24]. R2D2 and Loqs isoform D (Loqs-PD) have two type A dsRBDs, serving as partner proteins of *Dm*Dcr-2[24–26].

It is generally believed that two kinds of siRNA are generated by *Dm*Dcr-2, exo-siRNA for defense against viruses and endo-siRNAs (or esiRNAs) for protecting genome integrity from the serious threat of transposable elements[27]. *Dm*Dcr-2 has been observed in several conformational states during the cleavage of dsRNA substrates, including the translocation state, active dicing state, and post dicing state, among others[10,28]. The presence of Loqs-PD has been found to enhance the processing activity of *Dm*Dcr-2 by increasing the initial dsRNA-binding rates of its helicase domain[19,29,30]. After the dsRNA substrate cleavage, R2D2 helps *Dm*Dcr-2 to direct the product siRNA duplex into the RNA interference effector Ago2 with the Hsp70/90 chaperone systems, rather than enhancing its cleavage activity[7,31–34]. Therefore, R2D2 serves as an important component of the RISC loading complex (RLC). It has been reported that Loqs-PD and R2D2 act sequentially in the siRNA pathway, and they are required in a common silencing mean that is triggered by either exogenous or endogenous dsRNAs[35].

Like other cofactors, R2D2 and Loqs-PD interact with the Helicase domain of *Dm*Dcr-2 through their C-terminus. Recent studies have investigated the individual structures and functions of the *Dm*Dcr-2/Loqs-PD and *Dm*Dcr-2/R2D2 complexes, revealing that their binding sites do not overlap[10,36]. In addition, previous research has indicated the formation of a ternary complex involving *Dm*Dcr-2, Loqs-PD, and R2D2 in vivo[37]. However, the specific arrangement and synergistic interactions within this ternary complex remain poorly understood. In order to investigate the effects on *Dm*Dcr-2's function in the ternary complex during siRNA generation, we constructed the *Dm*Dcr-2 mutant (D1217N/D1476N, *Dm*Dcr-2[DDNN]) which exhibited no RNA dicing activity but possessed helicase activity in this work. Using cryo-electron microscopy (cryo-EM), we determined several structures of the protein complexes with dsRNAs. The structures reveal that R2D2 and Loqs-PD can bind to *Dm*Dcr-2 simultaneously and induce the complex to form an oligomer in the presence of dsRNAs and ATP. These findings provide insights into the interactions between the components.

## Results

### The architecture of the overall structures

We co-expressed and purified the *Dm*Dcr-2[DDNN] and wild-type R2D2 in insect SF9 cells and then incubated them with purified full-length Loqs-PD expressed in *E. coli*. The size-exclusion chromatography (SEC) results showed that the three components can form a stable complex in vitro (Supplementary Fig. 1a, b). To mimic the substrates and

products, we designed and transcribed two dsRNAs, one with a length of 50 bp and the other with a length of 19 bp (Fig. 1a). The protein complexes and RNAs were then incubated at equal molar ratios in buffer containing ATP and Magnesium ions. Then we use cryo-EM to analyze the structures of the ternary protein complex with 50-bp and 19-bp dsRNA substrates, respectively (termed 50-bp complex and 19-bp complex hereafter). Although large aggregates were not apparent in the original cryo-EM images (Supplementary Fig. 1c, f), all the 2D class averages revealed that protein-RNA complexes aggregate together and assemble into higher-order complexes (Supplementary Fig. 1d, h). From the 50-bp complex dataset, we acquired two structures: one in the dimer state with two *Dm*Dcr-2 molecules at a resolution of 3.72 Å, and another in the trimer state with three *Dm*Dcr-2 molecules at a resolution of 3.74 Å. In addition, from the 19-bp complex dataset, we obtained a trimer state structure with three *Dm*Dcr-2 molecules at a resolution of 3.70 Å. To facilitate clarity, we designated each monomer within the oligomer as 1st, 2nd, and 3rd from the bottom-up in the placement view (Fig. 1b–e). The protein components in both states displayed similar spatial arrangements, and the *Dm*Dcr-2 molecules were oriented in a similar direction. To our knowledge, this represents a novel organization of Dicer proteins (Fig. 1f). Notably, only the C-terminus region of R2D2 was well-defined in all the structures, serving as a scaffold between two adjacent *Dm*Dcr-2 molecules and indicating its crucial role in the formation and stabilization of these states. As in previous studies[10], a small segment of the C-terminus of Loqs-PD could be identified in our maps. Interestingly, this density was only present in the 1st monomer (Fig. 1b–e).

In the dimer state structure, the model of dsRNA could be well fitted into the continuous density; however, compared with free dsRNA, we observed significant distortion. The 1st and 2nd *Dm*Dcr-2 are linked by this dsRNA, and there are large angles of tilt and twist between the two Helicase domains (Supplementary Fig. 2a). This decoration pattern along dsRNA suggests that *Dm*Dcr-2's Helicase domain can adopt a head-to-tail arrangement. Previous results have shown that double-stranded RNA can induce *Dm*Dcr-2/Loqs-PD to form a stable dimer of the *Dm*Dcr-2/Loqs-PD/dsRNA complex with C2 symmetry[10]. In that complex, two *Dm*Dcr-2 monomers form a helicase-to-cap arrangement with the C-terminus of Loqs-PD located between two *Dm*Dcr-2 molecules to stabilize the overall conformation (Supplementary Fig. 2b). In the same study, the "mid-translocation state" structure captured the state in which two *Dm*Dcr-2 molecules bind back-to-back via their Helicase domains with one dsRNA. We further analyzed the dataset of *Dm*Dcr-2/Loqs-PD/50-bp-dsRNA/ATP used in that work and got a low-resolution structure where one monomer in dimer state was connected back-to-back to the third *Dm*Dcr-2 via their Helicase domains (Supplementary Fig. 2b–d). Interestingly, when R2D2 was introduced into the same system, the arrangement of the two *Dm*Dcr-2 monomers transitioned to a head-to-tail configuration (Supplementary Fig. 2e). This indicates that R2D2 plays a distinct role from Loqs-PD in the complex, as it can alter the aggregation mode of *Dm*Dcr-2.

After processing the 50-bp complex dataset, we identified a trimeric structure which can be fitted well with the trimeric structure obtained from the 19-bp complex dataset. The correlation value between the two structures was 0.9503, indicating that the structures are highly similar (Supplementary Fig. 3a). It is worth noting that the 50-bp dsRNA might not be long enough to connect all three monomers in the trimeric structure. We propose that the presence of the trimeric state is likely attributed to the use of palindromic sequence RNA in the sample. This RNA can form short hairpin dsRNA structures resembling 25-bp dsRNA (Fig. 1a). Therefore, the formation mechanism of the two trimers is similar. The structural comparison revealed that the arrangement of the *Dm*Dcr-2 molecules in the trimeric state is very similar to that of the dimeric state, but the RNA densities are discontinuous (Supplementary Fig. 3b, c). This discontinuity resulted in variations in the bonding details between adjacent monomers. While

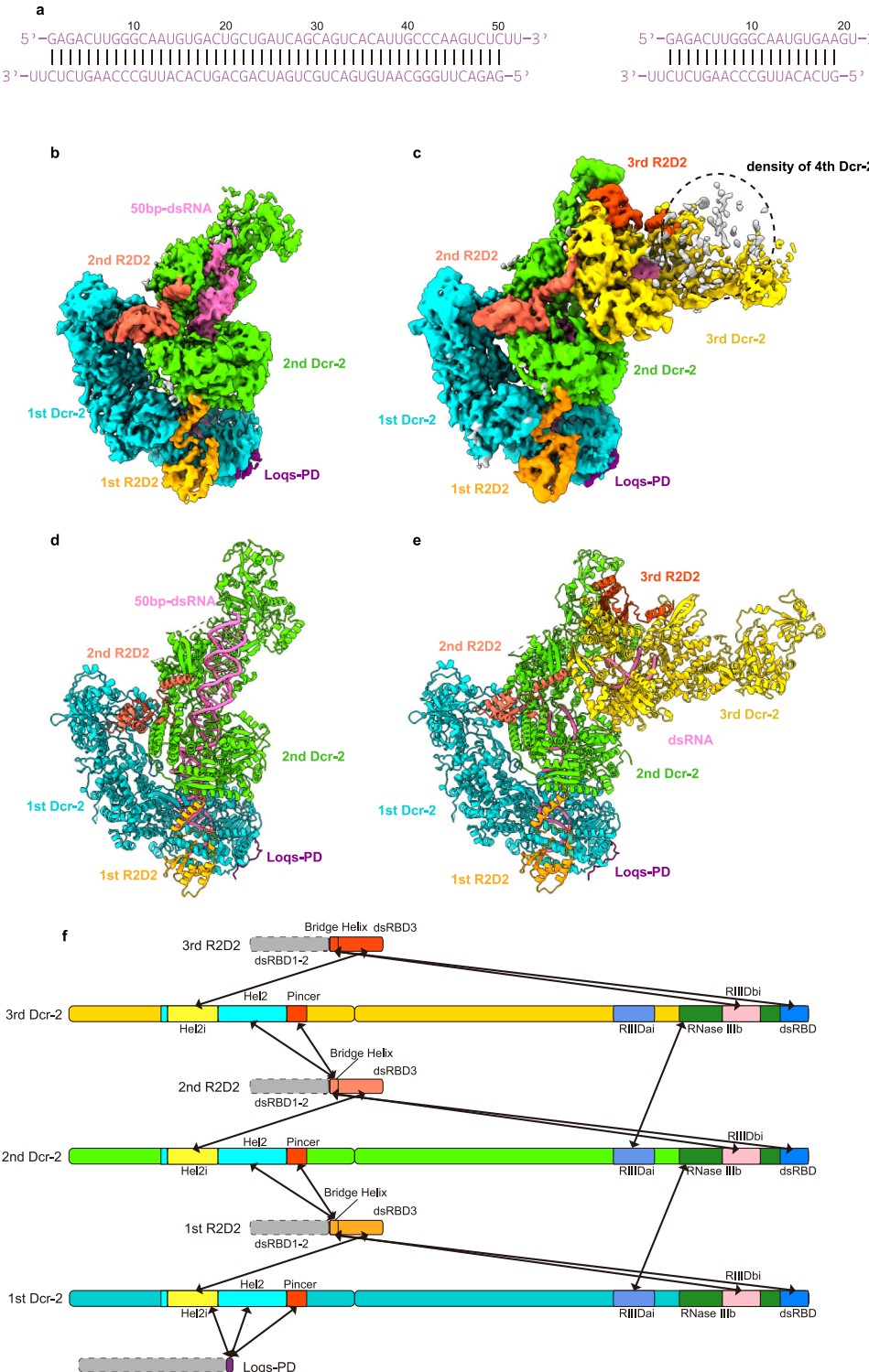

**Fig. 1 | Cryo-EM structures of *Dm*Dcr-2/R2D2/Loqs-PD in complex with 50-bp and 19-bp dsRNA. a** Schematic of 50-bp and 19-bp dsRNA. **b–e** The cryo-EM density map and atomic model of the complex in Dimer and Trimer state oligomer. **f** Color-coded domain architecture of the Trimer state oligomer. *Dm*Dcr-2, R2D2, and Loqs-PD are in the same color scheme is used as in (**e**). The intramolecular interactions are indicated as arrows. Domains participated in the intramolecular in *Dm*Dcr-2 are highlighted with Hel2i in yellow, Hel2 in cyan, Pincer in orange-red, RIIIDai in cornflower-blue, RIIIDb in forest-green, RIIIDbi in pink, and C-dsRBD in deep-sky-blue.

dsRNA played a supportive role in the dimeric state, the *Dm*Dcr-2 monomers in the trimeric state exhibited much closer proximity to each other. This difference is further supported by the relative distances and angles between adjacent *Dm*Dcr-2 monomers, which were highly consistent in the trimeric state (Supplementary Fig. 3d).

## R2D2 binds to the Hel2i of *Dm*Dcr-2, a similar binding position of TRBP on *Hs*Dicer

Although we used the full-length R2D2, only the C-terminal part (aa 186–311) of the structure could be well-modeled. This region consists of an α-helix and the C-terminal domain (Supplementary Fig. 4a, b).

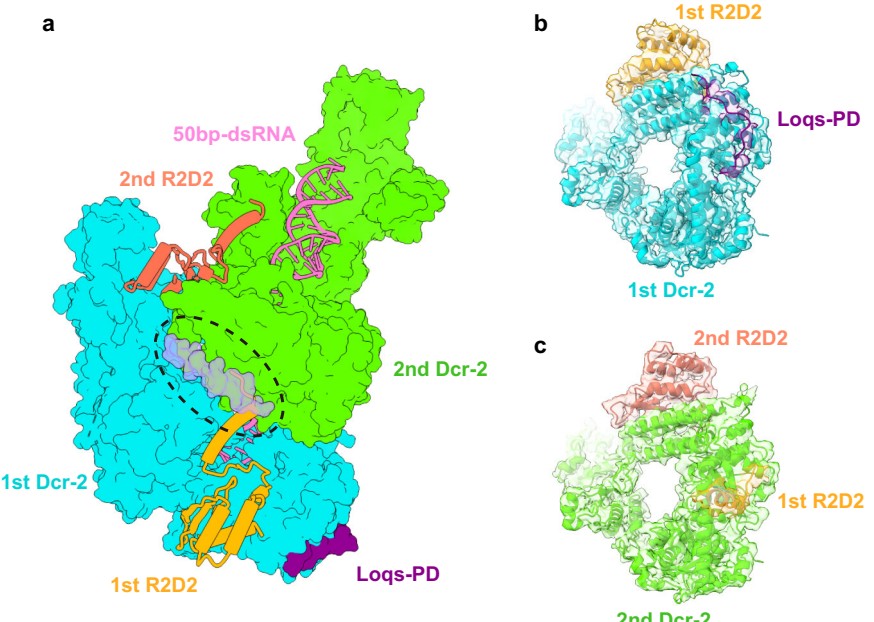

**Fig. 2 | Locations of R2D2 in the oligomers. a** Take Dimer state for example, R2D2 is located between the two *Dm*Dcr-2 monomers and inhibits the binding of the second *Dm*Dcr-2 to Loqs-PD. Two *Dm*Dcr-2 monomers are shown in surface mode. R2D2 and dsRNA are shown in cartoon mode. **b** Cryo-EM density and the fitted model of the Helicase domain of the 2nd *Dm*Dcr-2. The Loqs-PD binding position is occupied by R2D2. The cryo-EM map was shown at 70% transparency. **c** Cryo-EM density and the fitted model of the Helicase domain of the 1st *Dm*Dcr-2. The densities of Loqs-PD and R2D2 can be well-defined.

Despite the lack of sequence conservation, the C-terminal region adopts an α–β–β–β–α-fold and has a 3D structure similar to type B dsRBDs (Supplementary Fig. 4a). Besides, R2D2 binds to the Hel2i sub-domain of *Dm*Dcr-2 through this region, and both the binding position and binding mode are similar to those of TRBP on *Hs*Dicer[38] (Supplementary Fig. 4c). Therefore, it can be inferred that R2D2 is also composed of two type A dsRBDs and one type B dsRBD. The well-modeled region of R2D2 includes a distinct α-helix that acts as a bridge between the two Helicase domains, preventing the *Dm*Dcr-2 monomers from approaching each other (Fig. 2a). Since the sequence of this helix is not conserved in common cofactors (Supplementary Fig. 4a), we refer to it as the "bridge helix" and will discuss it further later.

In the 1st monomer of the quaternary complex of *Dm*Dcr-2/R2D2/Loqs-PD/19-bp siRNA, the presence of R2D2 does not affect the interaction of *Dm*Dcr-2 with Loqs-PD, as they bind to different sites. The structures show that R2D2 and Loqs-PD bind to the Helicase domain of the 1st *Dm*Dcr-2 together, indicating that their functions are not mutually exclusive. This is in contrast to the functions of TRBP and PACT on *Hs*Dicer, where they need to bind to the same position[10]. However, it is evident that only the 1st *Dm*Dcr-2 binds with Loqs-PD, while the same position in 2nd and 3rd *Dm*Dcr-2 is occupied by the "bridge helix" of R2D2 (Fig. 2a–c). Therefore, although R2D2 and Loqs-PD do not overlap in binding position on a single *Dm*Dcr-2, R2D2 can occupy the binding position of Loqs-PD in the neighboring *Dm*Dcr-2 when the quaternary complexes form an oligomer. This may represent a new form of restraint for R2D2 and Loqs-PD, as R2D2 bound to the first *Dm*Dcr-2 can inhibit the second *Dm*Dcr-2 from binding to Loqs-PD.

### Unique functions of the insertion regions in RNase-III domains of *Dm*Dcr-2

Within the RNase-III domain of proteins like Drosha or Dicer, there is often an insertion region of unknown function. Sequence alignment results show that both the sequences of insertion region of RNase IIIa (RIIIDai) and RNase IIIb (RIIIDbi) are not conserved[10]. The high-resolution structures of *Dm*Dcr-2 show that its RIIIDai consists of several α-helices (Supplementary Fig. 5a). Notably, this region is not

modeled in other solved structures of Dicer proteins[7-9,11,39], and the Alphafold2 prediction suggests that it is disordered[40]. Given its location at the interface of multiple oligomers (Fig. 3a, b), the RIIIDai may play important functional roles for *Dm*Dcr-2. This region contains many charged amino acids, so it may be essential for the formation and stabilization of different states by electrostatic interaction or hydrogen bonds (Fig. 3c, d).

During the process of dsRNA cleavage, the ATP-dependent *Dm*Dcr-2 adopts various conformations to ensure efficient and precise dicing. In comparison to previously reported structures[10], the state of *Dm*Dcr-2 monomer in the oligomer belongs to "translocation state". In this state, the Helicase domain binds to the dsRNA and translocates along it, powered by the hydrolysis of ATP, before the dsRNA terminus reaches the PAZ-Platform domains. (Fig. 3e). In this oligomeric state, the previously unstructured loop (aa 1551–1608) as part of the insertion region of RNase IIIb (RIIIDbi) can be partially built (while aa 1560–1593 is still missing) and is sandwiched between the "bridge helix" of R2D2 and *Dm*Dcr-2's C-terminal dsRBD (Fig. 3e, Supplementary Fig. 5b). The newly built loops (aa 1551–1559 and aa 1594–1608) form a hydrophobic surface with the dsRBD and interact with the "bridge helix" (Fig. 3f). Currently, the function of R2D2 is believed to serve as a component of RLC without enhancing the dicing activity. Compared to Loqs-PD, R2D2 has more interaction sites with *Dm*Dcr-2, which may stabilize the overall structure of *Dm*Dcr-2, thereby affecting the cleavage activity of *Dm*Dcr-2. To test this hypothesis, we analyzed the effects of Loqs-PD and R2D2 on *Dm*Dcr-2 cleavage activity. Cleavage assays show that R2D2 can weaken the dicing activity of *Dm*Dcr-2 in vitro (Fig. 3g and Supplementary Fig. 6). This may be attributed to the influence of R2D2 on the conformational changes that occur during *Dm*Dcr-2 dices its substrate.

### Factors that affect the assembly of oligomers

According to the structure of the oligomer, RIIIDai, RIIIDbi, and R2D2 are all potential stabilizers of the oligomer structure, as they are located at the interface of the oligomer. In order to demonstrate the effect of R2D2 on the oligomer, we introduced two constructs of R2D2:

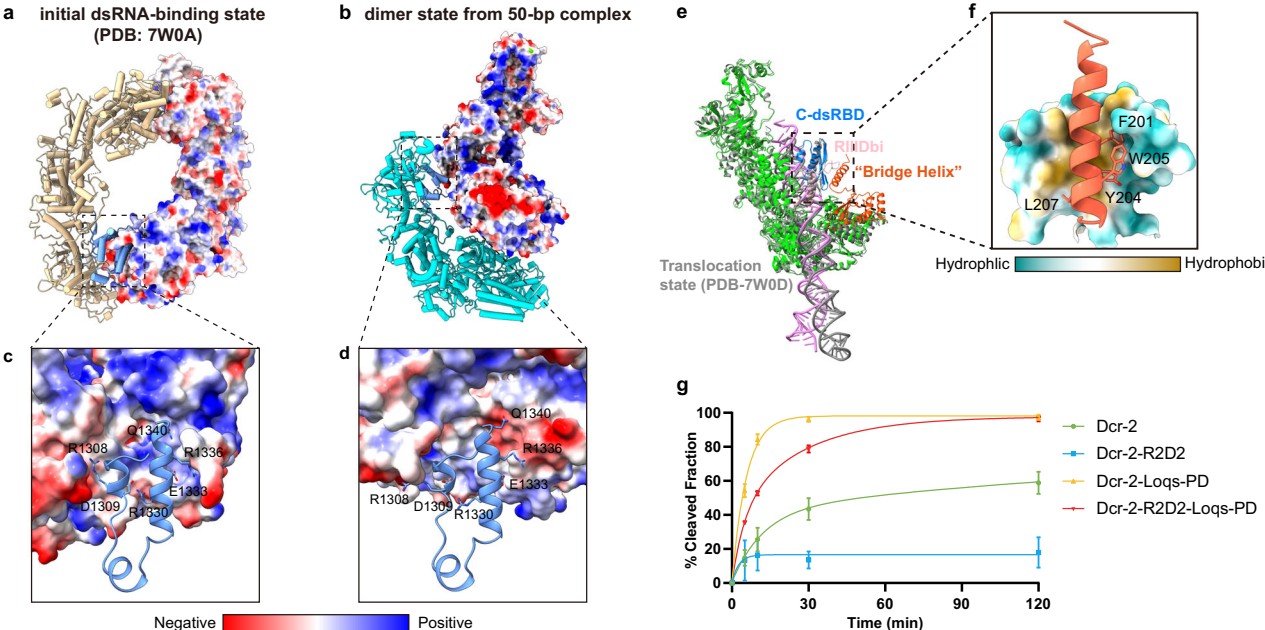

**Fig. 3 | The RNase DIIIa insertion regions of *Dm*Dcr-2 is located on the interface of different oligomers. a, b** The interface of PDB-7W0A and the Dimer state from 50-bp complex, one monomer shown in surface colored by electrostatic potential. The 1st monomer of *Dm*Dcr-2 in the Dimer state color with cyan, and another monomer in PDB-7W0A colored by tan. **c, d** Amino acid details at the interaction interface of boxed regions, partly charged or polar amino acids are labeled. **e** The *Dm*Dcr-2 monomer (lime) in the oligomer adopts the "translocation state" by comparing with PDB-7W0D (gray). R2D2 links the C-dsRBD and the Helicase domain of *Dm*Dcr-2, which may enhance the overall structural stability. **f** RIIIDbi forms a hydrophobic surface with C-terminal dsRBD of *Dm*Dcr-2 and binds to the "bridge helix" of R2D2. **g** The plot of cleavage efficiency of *Dm*Dcr-2, *Dm*Dcr-2/Loqs-PD, *Dm*Dcr-2/R2D2, *Dm*Dcr-2/R2D2/Loqs-PD with 106-bp blunt (BLT) termini dsRNA corresponding to Supplementary Fig. 6b (*n* = 3 each; means ± SD). Source data are provided as a Source Data file.

the sequence (aa 188–209) including the "bridge helix" replaced with 8*GS (R2D2$^{8*GS}$); or completely remove the "bridge helix" and its preceding region (R2D2$^{(215-311)}$). The BS3-cross-linked results showed that both the mutation and the truncation constructs significantly reduced the oligomerization rate of the complex (Fig. 4a, b and Supplementary Fig. 7a). Negative staining 2D averages showed that there were still some oligomers in R2D2$^{8*GS}$, but the high-resolution structure was not obtained from the more cryo-EM data, probably because the overall stiffness was reduced without this helix (Supplementary Fig. 7b, c). Moreover, R2D2$^{(215-311)}$ lost the ability to promote oligomer formation (Fig. 4b and Supplementary Fig. 7a), which also suggests that the two tandem dsRBDs of R2D2 might help initiate the oligomerization.

Environmental factors from the surrounding milieu can also influence oligomer formation. The negative staining EM results indicate that both dsRNA and ATP are required for oligomer formation, as shown by the 2D averages (Supplementary Fig. 7d, e). To investigate the oligomerization state under different conditions, we determined the ratio of oligomeric particles based on 2D averages of negative staining electron microscopy (Supplementary Fig. 8). The results show that ADP induces oligomer dissociation, while ATPγS maintains the oligomeric state (Fig. 4c). Therefore, after oligomer formation, *Dm*Dcr-2 can still hydrolyze ATP, and the proportion of oligomers gradually decreases with ATP consumption.

The oligomeric mode of trimeric state provides a further extension of the assembly mechanism that is not influenced by the length of dsRNA. In the structure of the 19-bp complex, we also observe some density of the fourth *Dm*Dcr-2 (Fig. 1c). Structural analysis indicates that the oligomeric mode of *Dm*Dcr-2 primarily depends on the dsRNA-induced conformational changes of the Helicase domain, independent of the length of dsRNA (19 bp or longer is sufficient). The oligomers can theoretically extend to form fibers, which is confirmed by negative stain electron microscopy (Fig. 4d and Supplementary Fig. 7a, d). To further analyze whether oligomerization exists in other arthropods, we compared the amino acid sequences of RIIIDai and

R2D2 in several insect species, as they located at the interface of the oligomer. The alignment results showed that RIIIDai of Dcr-2 and R2D2 are only conserved in Drosophilidae (Supplementary Fig. 9), suggesting that oligomerization of Dcr-2 may not be a common feature among insects.

## Discussion

We used cryo-EM to determine the structures of *Dm*Dcr-2 and its cofactors, Loqs-PD and R2D2, in complex with dsRNA of varying lengths in the presence of ATP and Mg$^{2+}$ ion. Surprisingly, we found that these components can form oligomers, enabling us to identify the fractions of R2D2 and Loqs-PD bound to *Dm*Dcr-2. The different binding sites revealed the compatibility relationship between R2D2 and Loqs-PD and their distinct effects on *Dm*Dcr-2. While the previous *Dm*Dcr-2/R2D2 dimer structure showed that dsRBD1-2 of R2D2 were positioned near the active dicing center mainly by the hydrophobic interactions between R2D2's dsRBD2 and *Dm*Dcr-2's central linker[36]. The C-terminal loop of Loqs-PD interact with *Dm*Dcr-2 turns out to give the dsRBDs of Loqs-PD a wider range for their stochastic movement and increase their dsRNA-binding probability[10,29]. It has been reported that Loqs-PD can promote the processing activity by increasing the initial dsRNA-binding rates of *Dm*Dcr-2's helicase domain instead of changing the processivity of *Dm*Dcr-2, suggested that Loqs-PD probably dissociated from dsRNA after the initial dsRNA-binding state formation[30]. Thus, the dsRBDs' density of Loqs-PD was missing in all reported structures. The multi-site interaction between R2D2 and *Dm*Dcr-2 limited the spatial location of R2D2[36], in contrast to the wider range of freedom of the two tandem dsRBDs of Loqs-PD. This difference is reflected in their respective functions, with Loqs-PD recruiting surrounding dsRNA substrates and loading them into the Helicase domain to accelerate dicing activity[30], while R2D2 is poised to capture product siRNA RNA duplex and load it onto Ago2. However, further investigation is needed to elucidate the inhibitory effect of R2D2 on *Dm*Dcr-2's dicing activity.

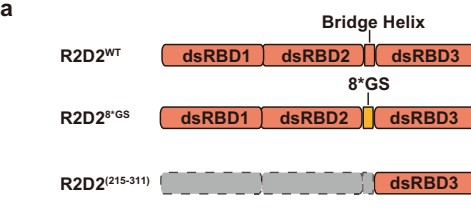

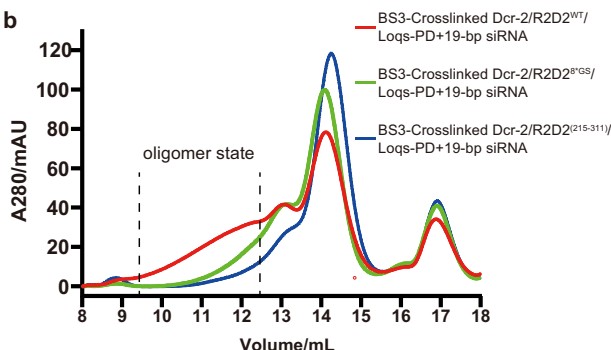

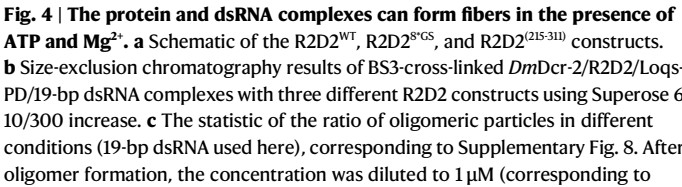

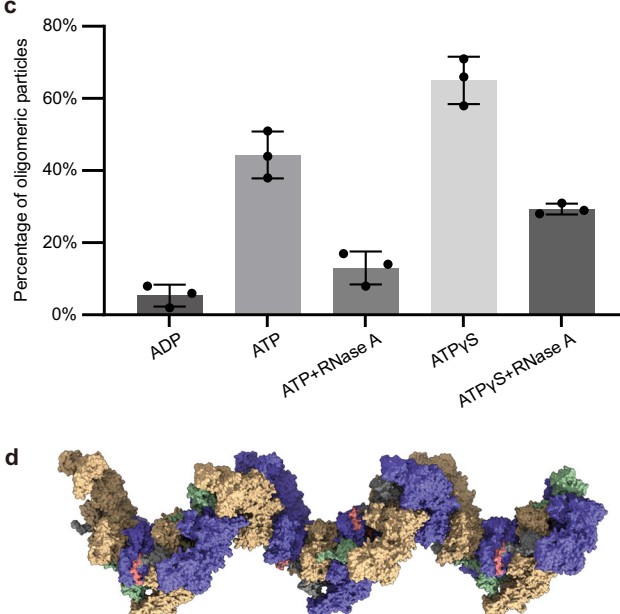

**Fig. 4 | The protein and dsRNA complexes can form fibers in the presence of ATP and Mg²⁺.** **a** Schematic of the R2D2^WT, R2D2^8*GS, and R2D2^(215-311) constructs. **b** Size-exclusion chromatography results of BS3-cross-linked *Dm*Dcr-2/R2D2/Loqs-PD/19-bp dsRNA complexes with three different R2D2 constructs using Superose 6 10/300 increase. **c** The statistic of the ratio of oligomeric particles in different conditions (19-bp dsRNA used here), corresponding to Supplementary Fig. 8. After oligomer formation, the concentration was diluted to 1 μM (corresponding to *Dm*Dcr-2/R2D2/Loqs-PD complexes) and then exposed to buffers containing final concentrations of 5 mM ADP, 5 mM ATP, 5 mM ATP with 0.1 μM RNase A, 5 mM ATPγS, 5 mM ATPγS with 0.1 μM RNase A, respectively. Negative staining samples were prepared after incubation at 25 °C for 1 h, and the ratio of oligomers was counted (*n* = 3 each; means ± SD). Source data are provided as a Source Data file. **d** The spring-like helical fiber model of the oligomer of the quaternary complexes based on the Trimer state structure.

The previously reported structure of the *Dm*Dcr-2/R2D2/siRNA complex demonstrated that two dsRNA duplexes bind to the Hel1 subdomain and R2D2[36]. However, in comparison to the apo state structure, the helicase domain remained unaltered (resembling the letter "C"). This implies that the mere presence of the dicing product dsRNA duplex is insufficient to induce conformational changes in the helicase domain of the *Dm*Dcr-2/R2D2 complex. Nevertheless, the structures of *Dm*Dcr-2 bound with both the 50-bp substrate and 19-bp duplex product demonstrated that these dsRNAs are incorporated into the helicase domain, prompting the transformation from a stretched C-shape to the activated O-shape state (Supplementary Fig. 10a–d). This conformational change is crucial both for *Dm*Dcr-2 to process its substrates and for oligomer formation (Fig. 5). Although only a small proportion of Loqs-PD was identified at the periphery of all the oligomers, it is highly probable that it plays an important role in the proper assembly of this oligomer. The absence of Loqs-PD resulted in only a small percentage of *Dm*Dcr-2/R2D2 with dsRNA complexes forming oligomers when following the same experimental procedure. However, due to the low oligomeric ratio and the dominant orientation problem, the three-dimensional structure of the complex could not be obtained (Supplementary Fig. 10e). Therefore, the synergistic functions of Loqs-PD and R2D2 promote the formation of oligomers.

Previous study shows that *Dm*Dcr-2 and R2D2 are co-localized in vivo[37]. Our fluorescence results also support the co-localization of *Dm*Dcr-2, R2D2, and Loqs-PD in S2 cells, suggesting the presence of a ternary complex and oligomers in vivo (Supplementary Fig. 11). Understanding the biological function of *Dm*Dcr-2 oligomerization is of great significance. Based on our structural analysis, we propose several hypotheses regarding its potential functions. The trimeric structure of the complex allows for the full loading of siRNA duplexes into the Helicase domains, providing temporary protection against degradation until Ago2 is in close proximity. Although the loading of these siRNA duplexes onto Ago2 could not be confirmed, the addition of RNaseA decreased the ratio of oligomers in the presence of ATP compared to ATPγS, suggesting that the siRNA duplexes could be released (Fig. 4c). Oligomer formation restricts the function of Loqs-PD through R2D2, ensuring that released siRNA is loaded onto R2D2 and subsequently onto Ago2, rather than being captured by Loqs-PD and the Helicase domain (Fig. 5). The dependence of oligomerization on dsRNA and ATP implies that this phenomenon may exist in vivo. While attempts to observe the presence of these oligomers in the S2 cell line were unsuccessful. Future research will be needed to investigate the role of these oligomers in vivo, especially their unique functions for *Drosophila*.

## Methods

### Protein purification and dsRNA preparation

Plasmids construction, protein expression and purification of full-length *Dm*Dcr-2, *Dm*Dcr-2^DDNN, Loqs-PD were same as described previously[10]. Briefly, *Dm*Dcr-2 or *Dm*Dcr-2^DDNN were expressed using the Bac-to-Bac baculovirus expression system (Invitrogen) in sf9 cells. The collected cells were resuspended in buffer A (150 mM NaCl, 20 mM Tris-HCl pH 8.0, 10% glycerol, 20 mM imidazole) with 0.5 mM PMSF and protease inhibitors, and lysed by adding 0.5% Triton X-100 and shaken gently for 30 min at 4 °C. *Dm*Dcr-2 or *Dm*Dcr-2^DDNN were purified to homogeneity using Ni-NTA affinity, Hitrap Q column (Cytiva), 2nd Ni-NTA affinity and size-exclusion chromatography using the Superdex 200 10/300 Increase column (Cytiva) (in that order). The peak fractions corresponding to target proteins were collected and concentrated to about 10 mg/ml (50 μM) and stored at −80 °C in SEC buffer (100 mM NaCl, 20 mM Tris-HCl pH 8.0, 1 mM DTT). Loqs-PD was expressed in *Escherichia coli* BL21 (DE3). Loqs-PD was first purified by Ni-NTA affinity chromatography. Using protease ULP1 to remove the 6×His−SUMO tag, and dialysis was applied to remove imidazole. The

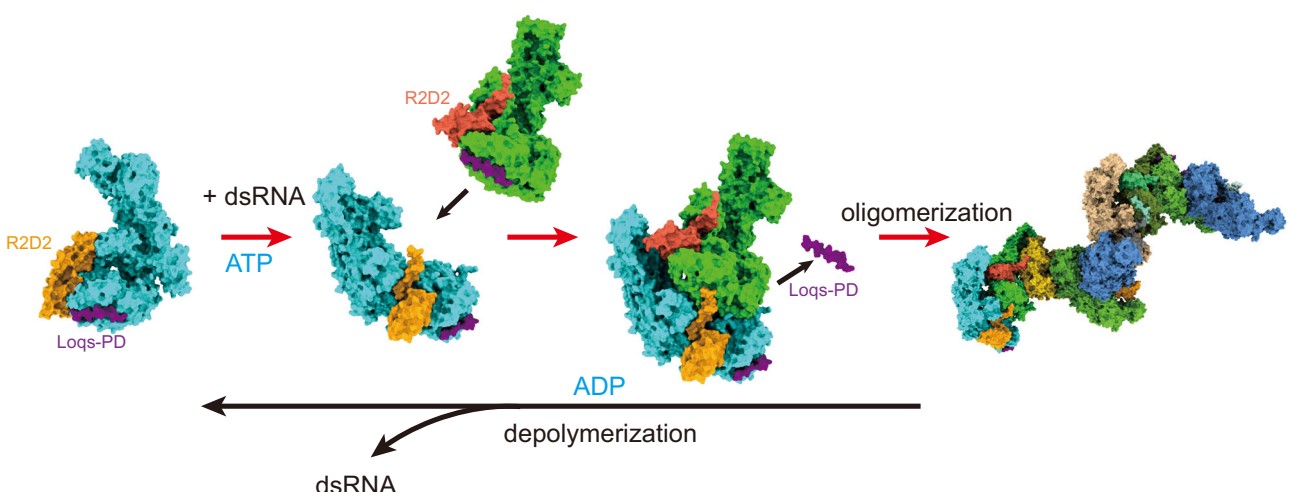

**Fig. 5 | Model of R2D2 and Loqs-PD mediate the formation of *Dm*Dcr-2 oligomers.** Loqs-PD can recruit dsRNA substrates from the surrounding environment and then load them into the Helicase domain of *Dm*Dcr-2. This process triggers conformational changes in the Helicase domains, which ultimately promote oligomerization. Once oligomerization occurs, Loqs-PD is released. While each of these steps is reversible, the oligomers tend to depolymerize when the ATP concentration becomes low.

sample was then applied to a second Ni-NTA chromatography and the flow-through was collected for size-exclusion chromatography using the Superdex 200 16/600 column (Cytiva). Fractions corresponding to Loqs-PD were collected and concentrated to about 10 mg/ml (250 μM) and stored at −80 °C in buffer containing 500 mM NaCl, 20 mM Tris-HCl pH 8.0, and 1 mM DTT.

Full-length R2D2 (UniProtKB: Q9VLW8) was PCR amplified from *Drosophila* cDNA and cloned into pFastBac (with 6×HIS affinity tag). The other mutations were generated using a standard PCR-based cloning strategy and cloned into the same vectors. All plasmids' identities were confirmed by sequencing analysis. Expression and purification of *Dm*Dcr-2/R2D2 and *Dm*Dcr-2/R2D2-mutions were followed the same protocol. Take *Dm*Dcr-2/R2D2, for example, *Dm*Dcr-2/R2D2 was co-expressed using the Bac-to-Bac baculovirus expression system (Invitrogen) in sf9 cells at 27 °C. One liter of cells (2 × 10⁶ cells per ml, medium from Sino Biological) was infected with 20 ml baculovirus at 27 °C. After growth at 27 °C for 48 h, the cells were collected, resuspended in buffer A with 0.5 mM PMSF and protease inhibitors, and lysed by adding 0.5% Triton X-100 and shaken gently for 30 min at 4 °C. *Dm*Dcr-2/R2D2 was purified to homogeneity using Ni-NTA affinity (GE Healthcare), anion exchange (Hitrap Q column, Cytiva), 2nd Ni-NTA affinity, and size-exclusion chromatography (Superdex 200 10/300 Increase column, Cytiva). Fractions corresponding to target complexes were collected and concentrated to about 10 mg/ml (42 μM) and stored at −80 °C in SEC buffer.

Purified *Dm*Dcr-2 ᴰᴰᴺᴺ/R2D2 and Loqs-PD were mixed at a molar ratio of 1:1.5 and incubated on ice for 1 h before the size-exclusion chromatography (Superose 6 10/300 Increase column, Cytiva) equilibrated with SEC buffer. Fractions corresponding to a three-component complex were collected for further experiments.

### Preparation of dsRNAs
The sense and antisense strands of 19-bp siRNA were ordered from Dharmacon. The 50-bp and 106-bp dsRNAs were in vitro transcribed using T7 RNA polymerase. The pUC19 plasmids containing target sequences with 3′-HDV ribozyme sequences were linearized by EcoRI, extracted with phenol–chloroform, and precipitated with isopropanol. The in vitro transcription reaction was performed at 37 °C for 5 h in the buffer containing 100 mM HEPES-K (pH 7.9), 10 mM MgCl₂, 10 mM DTT, 6 mM NTP each, 2 mM spermidine, 200 μg/ml linearized plasmid and 100 μg/ml T7 RNA polymerase. The transcription products were purified by 8% denaturing urea PAGE, eluted from gel slices, and

precipitated with isopropanol. After centrifugation, the RNA precipitant was collected, washed twice with 70% ethanol and air-dried, and the RNA was dissolved in ultrapure water. We next used T4 PNK (NEB, M0201) to remove the 2′,3′ cyclic phosphate at the 3′ end of the RNA. The sense and antisense strands of 19-bp and 106-bp dsRNA were annealed by heated to 95 °C for 3 min and then slowly cooled to room temperature in the buffer containing 50 mM HEPES-Na pH 8.0, 100 mM NaCl, 1 mM MgCl₂.

### In vitro dsRNA cleavage assays
*Dm*Dcr-2 and its cofactor complexes cleavage assays were performed in cleavage assay buffer (100 mM Tris-HCl pH 8.0, 100 mM NaCl, 1 mM DTT) with an equal molar ratio of unlabeled dsRNA (1.2 μM protein and 1.2 μM dsRNA). Protein complexes and dsRNA were preincubated at 25 °C for 15 min, then added ATP and MgCl₂ to a final concentration of 5 mM to start the reaction. The reactions were stopped with an equal volume of 2× formamide loading buffer (95% formamide, 20 mM EDTA, 0.1% SDS, 0.005% xylene cyanol, 0.005% bromophenol blue). Samples were separated by 12% denaturing PAGE, post-stained with Gelred, and visualized on Typhoon FLA-9000 (GE Healthcare) system.

### Size-exclusion chromatography after BS3-crosslinking
In all, 900 pmol of the purified Dcr-2/R2D2/Loqs-PD complexes with 1080 pmol 19-bp siRNA were preincubated in reaction buffer containing 50 mM HEPES-Na pH 8.0, 100 mM NaCl 5 mM ATP, 5 mM MgCl₂ and 5% glycerol at 25 °C for 1 h. Then added 1 mM bis(sulfosuccinimidyl)-suberate (BS3; Thermo Fisher Scientific, 21580) and incubated at 25 °C for 2 h and quenched by 20 mM Glycine. Cross-linked complexes were analyzed by negative staining EM and size-exclusion chromatography using Superose 6 10/300 increase column.

### Negative staining EM data collection
The *Dm*Dcr-2/R2D2/Loqs-PD complex with dsRNA samples were diluted to a final concentration of ~100 nM (corresponding to *Dm*Dcr-2/R2D2/Loqs-PD complexes) using SEC buffer (with additional 5 mM ATP and 5 mM MgCl₂ if needed) immediately before the negative-stained specimen was prepared. In total, 4 μl of the solution was applied to a glow-discharged holey carbon grid with a thin layer of continuous carbon over the holes. After adsorption on the grid for 1 min, the grid was negatively stained with three droplets of 2% (w/v) uranyl acetate solution for total of 1 min, the residual stain was blotted off, and the grid was air-dried. The specimen prepared as above was inserted into

an FEI F20 transmission electron microscope equipped with a FEG and operated at 200 kV acceleration voltage for examination. We examined the specimen and collected digital micrographs at a nominal magnification of 50,000× with a total dose of about 20 e⁻/Å² and a defocus value ranging from −2.0 to −1.5 µm on a Gatan Ultra4000 CCD camera, with a final pixel size of 2.2 Å.

### Image processing of negative staining
All the image processing was performed in CryoSPARC[41]. For each kind of samples, 20 micrographs were collected and imported into CryoS-PARC. As many particles as possible were picked automatically using blob picker. Particles were extracted using a box size of 200 × 200 and then resized to 100 × 100. Then two rounds of reference-free 2D classification were performed. Contaminants were removed for the first round, and the ratio of oligomers is determined from the two-dimensional averaged images of the second round.

### Immunofluorescence staining
The S2 cells were cultured in Schneider Drosophila Medium with 10% fetal bovine serum (FBS). Transfections were performed with Fugene HD (Promega) according to the manufacturer's instructions. The fluorescent protein-tagged *Dm*Dcr-2/Loqs-PD/R2D2 were constructed in the pUAST vector. All DNA plasmids were cleared of endotoxin. For living cell imaging experiment, 1 µg of plasmids for each protein were co-transfected in a 35-mm dish with a glass bottom (Nest, 801001). Cells were washed twice with PBS and fixed in 500 µL 4% Paraformaldehyde (PFA, sigma, F8775). After that, cells were incubated in permeabilization buffer (0.1% Triton X-100) in PBS for 20 min at room temperature.

### Live-cell imaging
S2 cell line was transiently transfected with fluorescent protein-tagged *Dm*Dcr-2/R2D2/Loqs-PD for 48hrs and imaged on an Olympus FV3000 confocal microscope using a ×60 oil objective.

### Cryo-EM sample preparation and data collection
Protein complex and two kinds of dsRNA were mixed at a molar concentration of 1:1, respectively, and incubated at 25 °C for 2 h before sample preparation. For each sample, an aliquot of 4 µl of purified samples was diluted to 1 µM using SEC buffer with an additional 5 mM ATP and 5 mM MgCl₂ and applied to a reduced graphene grid[42] (Shuimu BioSciences Ltd.), which were glow-discharged (in a HARRICK PLASMA), for 10 s at middle level after 2 min evacuation. The grids were then blotted by 55 mm filter paper (Ted Pella) for 0.5 s at 22 °C and 100% humidity, then flash-frozen in liquid ethane using FEI Vitrobot Marke IV. Cryo-EM data were collected on different Titan Krios electron microscopes, all of which were operated at 300 kV, equipped with a Gatan K3 direct electron detector and a Gatan Quantum energy filter. Data were automatically recorded using AutoEMation[43] (50-bp dsRNA dataset) or EPU (19-bp dsRNA dataset) in counting mode and defocus values varied from −1.5 to −2.0 µm. For other parameters of the two datasets, see Supplementary Table 1.

### Image processing and 3D reconstruction
For different cryo-EM datasets, the image processing was adopted in similar steps. All the raw dose-fractionated image stacks were two times Fourier binned, aligned, dose-weighted, and summed using MotionCor2[44]. The contrast transfer function (CTF) parameters were estimated using CTFFIND4[45]. Particles were picked automatically using blob picker in CryoSPARC[41]. After one round of reference-free 2D classification, and several rounds of 3D reference-based classification in RELION[46], using the initial 3D reference models obtained by Ab-initio reconstruction in CryoSPARC, particles from good 3D classes, with better overall structure features, were selected for 3D refinement. The final high-resolution homogeneous refinement and local resolution

distribution were performed in CryoSPARC. The resolutions were determined by gold-standard Fourier shell correlation. For the details of each dataset, please see Supplementary Fig. 1.

### Model building and refinement
The initial atomic model of R2D2 was predicted using Alphafold2. *Dm*Dcr-2, Loqs-PD, and dsRNA were separated from PDB-7W0A. The models were docked into EM density maps in UCSF ChimeraX[47] and then adjusted manually in COOT[48]. Finally, all the models were refined against the EM map by PHENIX[49] in real space with secondary structure and geometry restraints. All figures in the manuscripts were illustrated by UCSF ChimeraX[47] and UCSF Chimera[50]. The final models were validated in PHENIX software package. The model statistics are summarized in Supplementary Table 1.

### Reporting summary
Further information on research design is available in the Nature Portfolio Reporting Summary linked to this article.

## Data availability
All the cryo-EM maps and models have been deposited in the wwPDB OneDep System. The EMD accession codes of Dimer state from 50-bp-oligomer, Trimer state 19-bp-oligomer, and Trimer state 50-bp-oligomer are 34707, 34708, 34709. The PDB ID codes of Dimer state from 50-bp-oligomer, Trimer state 19-bp-oligomer are 8HF0 and 8HF1, respectively. Other parameters are listed in Supplementary Table 1. Other structural models cited in this study for analysis (7W0A, 7W0D, 4WYQ, and 7V6C) are also accessible through the PDB. All other data or materials can be obtained from the corresponding author upon request. Source data are provided with this paper.

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

## Acknowledgements

We thank Hong-Wei Wang at Tsinghua University for the support, guidance, and suggestion on this work. We thank Qinghua Liu from the National Institute of Biological Sciences, Beijing for the *Dm*Dcr-2 plasmid. We would like to thank Yan Dong's lab for providing the S2 cell line. We thank Jianlin Lei, Xiaomin Li, and Fan Yang at Tsinghua University for assistance of data collection. We acknowledge the Tsinghua University Branch of the China National Center for Protein Sciences (Beijing) and Shuimu BioSciences Ltd. for providing the cryo-EM facility support and computational facility support. This research is supported by the National Key Research and Development Program of China (2018YFC1003800 to J.M.) and the National Natural Science Foundation of China (31971130 and 31230041 to J.M., 32000849 to J.W.).

## Author contributions

J.M. and J.W. conceived the study. J.H. and X.Y. initiated the project. T.D., S.S., J.W., and J.M. designed experiments. T.D. and S.S. prepared the samples and performed the biochemical experiments. J.W. and S.S. performed cryo-EM experiments and structure determination. J.W. and S.S. built and refined the models. T.D., S.S., J.W., Y.H., and J.M. analyzed the data, T.D., S.S., J.W., and J.M. wrote the manuscript with input from other authors.

## Competing interests

The authors declare no competing interests.
