## [Peer Review File · Nature Communications]

Structural mechanism of R2D2 and Loqs-PD synergistic modulation on DmDcr-2 oligomersREVIEWER COMMENTS

Reviewer #1 (Remarks to the Author):

Manuscript by Deng and colleagues describes cryo-EM structures of Dm Dicer-2 with R2D2, Loqs-PD and two dsRNA helices (50- and 19-bp long). Understanding the roles of R2D2 and Loqs-PD in Dicer-2 function of dsRNA cleavage and product loading onto Ago2 has been the rationale for this work and has not been addressed by previous cryo-EM studies. Surprisingly, the authors report that DmDcr2 complexed with dsRNA and R2D2/Loqs-PD undergo di- or tri-merization, and FRAP experiments suggest high stability of the oligomers.

While the structures of DmDcr-2 complexes with R2D2, Loqs-PD (only the C-terminus of which is resolved) and dsRNA could provide interesting mechanistic insights into Dcr-2 function and the observation of oligomers is intriguing, the manuscript is disappointingly shallow as it lacks a deeper analysis of mechanistic implications into the roles of R2D2, Loqs-PD (and comparison with other Loqs-proteins and Dcr-1) and the role of oligomerization. A large body of work is neglected, e.g. the biochemical data discussing the possible roles of Dcr2 oligomerization vs. processivity on long dsRNA (for example, see <https://www.ncbi.nlm.nih.gov/pmc/articles/PMC3061311/> and <https://www.ncbi.nlm.nih.gov/pmc/articles/PMC3115569/> as well as subsequent manuscripts by these and other groups). As such, the manuscript needs to be substantially revamped to address the findings in the context of the existing literature. Additional detailed criticisms below must be addressed to improve the revised manuscript.

1. Cryo-EM structures of Dcr-2 with R2D2 (<https://pubmed.ncbi.nlm.nih.gov/35768503/>) and with Loqs-PD (<https://www.nature.com/articles/s41586-022-04911-x>) were recently reported. Deng et al seem to argue that oligomerization of Dcr-2 requires both R2D2 and Loqs-PD. But Loqs-PD is bound at the periphery of Dcr-2 and is not engaged in oligomer formation. Why does the current study observe Dcr-2 oligomers as opposed to the Dcr-2*R2D2 structure? What are the functional implications of both studies (this paper and Yamaguchi et al) and the differences observed between the studies?
2. Only the Hel2-binding region of Loqs-PD was observed in this and previous work. The recent Dcr-1*Loqs-PB structure finds all three domains of Loqs-PB interacting with the Dicer and RNA substrate (<https://pubmed.ncbi.nlm.nih.gov/36182693/>). Discuss functional/structural differences between Loqs-PB and Loqs-PD that might lead to the lack of Loqs-PD interactions with dsRNA or Dcr-2.
3. Line 109: "Same as the previous structures" – which ones? Change to "As in previous studies (references), only the small segment ... is defined in our maps...".
4. Line 159: "R2D2 and Loqs-PD bind to the Helicase domain of the 1st DmDcr-2 at the same time..." – This must be a mistake, as the timing of their binding is unknown from this work. The proteins likely bind sequentially rather than simultaneously.
5. Line 171 "this region is missing in all the other solved structures ...". Do you mean that the region was not modeled in the structures? Or that the sequence of the region is missing?
6. Lines 178-190. In this paragraph, the authors discuss the "translocation" and "active dicing" states. How are these states defined structurally? An introduction explaining the structural differences, and why they are interpreted as these functional states, would be helpful.
7. Line 185 "Two loop forms" – unclear.

8. From FRAP experiments, it is not clear how the roles of R2D2, Loqs-PD and dsRNA were tested in oligomerization of Dcr-2. FRAP experiments with individual cofactors, and with their combinations must be performed to address this point.

9. In Discussion, the authors suddenly talk about the “C-shape” and “O-shape” states. It would help the reader if the functional meaning and the structural differences are discussed.

10. Misleading jargon (like “Loqs-PD grabs the surrounding RNA”) should be avoided. Most structural mechanisms involve stochastic interactions and stochastic movements, rather than directed purpose-driven actions (such as “grabbing”). Furthermore, the authors do not demonstrate any evidence that Loqs-PD interacts with RNA. Improve the discussion by considering that interactions are stochastic; discussion should include the analysis of the data in this work and in previous genetic, biophysical, biochemical and structural studies.

Reviewer #2 (Remarks to the Author):

Drosophila Dicer2 produces siRNA to prevent viral infection and protect genome integrity. Two cofactors, R2D2 and Loqs-PD, bind to Dicer2 to aid processing the dsRNA substrate and loading the siRNA product to Ago2, respectively. The authors solved the structures of the Dicer2-Loqs-PD complex and revealed how the dsRNA substrate are recognized, translocated, and processed in the previous study. Others solved the structures of the Dicer2-R2D2 complex and explained the mechanism underlying the strand selection of the siRNA product. In this study, the authors determined the structures of Dicer2 in the presence of both Loqs-PD and R2D2. The major finding is that Loqs-PD induces R2D2-mediated oligomerization of Dicer2. High resolution view of the specific interactions of two cofactors with Dicer2 suggests evolutionary designed functions of two cofactors, which has been unknown so far. The oligomerization of Dicer proteins by cofactors has never been shown and is an interesting feature of *Drosophila* Dicer2. However, it is unknown what is the advantage of the oligomerization or whether the oligomerization state exists in vivo. Therefore, the physiological importance of the oligomerization and thus its detailed mechanism presented in the manuscript is unclear. While the oligomeric structure is interesting and potentially important, more extensive structural analyses including detailed comparison with previous structures of Dicer2 are recommended to make hypotheses and conclusions reasonable and understandable. In addition, there are too many typos and grammar errors. A professional English editing service is highly recommended.

The major points

1. The introduction is not sufficiently provided to fully understand the presented structures and discussions. For example, the 'translocation state' has never been explained in this manuscript. Without previous knowledge, one could easily think it is related to the cellular localization of Dicer2, not the translocation of RNA in the protein complex. The previous work by the authors and others need to be described in details for broad readership.

2. The authors need to provide at least one evidence that shows physiological importance of the oligomerization. For examples, one could observe the oligomeric state of Dicer2 in cells or tissues or investigate the effect of a mutation in R2D2 that prevent the Dicer oligomerization on the processing activity and selectivity in cells.

3. Without RNA, what was the structure of the protein complex like? Was it the 1:1:1 complex of Dicer2, R2D2, and Loqs-PD? If this is true, after the incubation with RNA, the majority of Loqs-PD proteins should have been dissociated. Then, how was Loqs-PD dissociated? Simply by the difference of the binding affinity? The authors need to include discussion about how the oligomeric state is formed during the cryo-EM sample preparation.

4. The binding region of 50-bp-RNA in Dicer2 is overlapped with the oligomeric interaction site. Then, how could higher order oligomers be formed as shown in Fig 4a. Is this also related to the smaller size of the oligomer in the presence of 50-bp-RNA compared with that in the presence of 19-bp-RNA?

5. The authors mentioned the the oligomerization may increase the processing efficiency by increasing local protein concentration. However, the oligomerization appeared to stop the processing cycle and hold Dicer at the translocation conformation. Then how can the oligomerization increase the efficiency in the oligomeric state? The discussion seems contradictory.

6. There are many typos and grammer errors. A professional english editing service is highly recommended.

The minor points

Line 31. It is difficult to understand the connection between large oligomers and a common evolutionary origin. Does this mean that many proteins in RNA-induced innate immunity oligomerize? Then this needs to be more properly described.

Line 58 and 60. The functions of TRBP, R2D2, and Loqs proteins needs to be explained before these sentences. No description on their functions is included in the introduction at all.

Line 64. The features of the substrate of Dicer2 and how they are processed need to be described before this section.

Line 71. The author described "it was reported that Loqs-PD and R2D2 act sequentially in the siRNA pathway". How solid would be this data? What would be still unclear? Since the presented work showed they can act simultaneously, the authors could raise some questions from this previous data.

Line 78. References need to be cited.

Line 95. This sentence can be mistaken for that protein:ATP is 1:1.

Line 106. Loqs-PD and R2D2 have large regions missing in the cryo-EM map. These need to be more clearly described in the text and shown in Fig 1f (e.g. by representing the missing part as gray color)

Line 129. How the different role would be possible needs to be explained here based on the structural comparisons.

Line 131. This paragraph is confusing, since there was no description that the trimeric structure was solved for both 19bp- and 50bp-dsRNA. Line 100 misleads the readers to understanding that only one structure for each dsRNA substrate was solved.

Line 139. Trimer need to be specified to the trimer with 50-bp-RNA.

Line 141. The sentence is difficult to understand. Please rephrase it.

Line 159. Do they really bind at the same "time"? "together" may be better.

Line 165. What restraint did the authors mean?

Line 168. Please give an information that RNase IIIs have two long insertions before this

Line 180. "translocation state" needs to be defined somewhere before this sentence.

Line 191. This hypothesis seems easy to test. The authors already have all materials.

Line 200. How is the complex labeled? by Alexa488 antibody or NHS? No information is found even in Methods.

Line 223. Could this be explained by the previous structures of the Dicer2-Loqs-PD complex?

Line 227. Is it confirmed that Loqs-PD initializes the oligomerization process? It seems that dsRNA, not Loqs-PD, initializes the oligomerization, based on the sample preparation protocol.

Line 240-243. The discussion is difficult to understand. More proper phrasing and background are required.

Line 323. The specific information of resins and columns need to be provided. Especially, the size of the protein complex cannot be estimated due to no information of SEC column.

Fig 3 legend includes an incomplete sentence.

Extended Data Fig. 5 legend. The alignment contain three green boxes. Which is the insertion mentioned

in Line 168?

Extended Data Fig. 1a. Reference molecular weights need to be indicated to estimate the sizes of the protein peaks.

Extended Data Fig. 6 is difficult to see how the dicer conformation and the R2D2 location change. The figure sizes need to be increased. The structure of 7V6C need to be colored by domains and proteins.

Reviewer #3 (Remarks to the Author):

In this manuscript (MS), Deng and colleagues solved the structures of DmDcr-2/R2D2/Loqs-PD/dsRNA using the Cry-EM approach. They found a dimer of DmDcr-2/R2D2 and one Loqs-PD on 50-bp dsRNA and a trimer of DmDcr-2/R2D2 and one Loqs-PD on 19-bp dsRNA. Although the cryo-EM structure of DmDcr-2/R2D2 and DmDcr-2/Loqs-PD were already solved, the structure of the ternary complex of DmDcr-2/R2D2/Loqs-PD in dsRNA solved in this study is a great addition to our understanding of complex assembly involving DmDcr-2 and its cofactors. The structures are at high resolution, and the description and interpretation of the structures are accurate and comprehensive. However, I have several major comments below.

Major points

1. The study did not address functions of the oligomer states of the DmDcr-2/R2D2/Loqs-PD in the DmDcr-2 processing mechanism and other cellular functions of DmDcr-2. Why does the enzyme complex need to be dimerized on the substrate, 50-bp dsRNA? Why does the enzyme complex need to be trimerized on the cleaved product, 19-bp dsRNA?
2. What are the cells' oligomer states of DmDcr-2/R2D2/Loqs-PD?
3. In the Extended Data Fig. 1, R2D2, smaller than Loqs PD, had a higher density in the gel than Loqs-PD. This indicated that the assembled DmDcr-2/R2D2/Loqs-PD already contained more DmDcr-2/R2D2 than Loqs-PD. What is the oligomer state of DmDcr-2/R2D2/Loqs-PD without RNAs?
4. In figure 4, they found that DmDcr-2/R2D2/Loqs-PD/dsRNA complexes can form huge irregular oligomers in the presence of ATP and Mg⁺⁺ in vitro. 1) Are the dimeric and trimeric states of the complexes shown in Fig. 1 just a small portion of the various oligomer state? 2) What are the oligomer states of DmDcr-2/R2D2/Loqs-PD complexes without RNA in the presence of ATP and Mg⁺⁺ in vitro? Whether RNAs induced the complex oligomerization?
5. They claimed that "R2D2 and Loqs-PD can bind to different regions of DmDcr-2 without spatial clash". How could they always find only one Loqs-PD in either dimeric or trimeric DmDcr-2 containing 2 (or 3) of R2D2 + 2 (or 3) of DmDcr-2?
6. Since the authors obtained DmDcr-2/R2D2, DmDcr-2/Loqs-PD, and DmDcr-2/R2D2/Loqs-PD complexes and they claimed that the structures they received were not DmDcr-2 in "active dicing" state. And they thought that R2D2 traps DmDcr-2 in the "translocation state." The authors should compare the dicing activity of DmDcr-2/R2D2, DmDcr-2/Loqs-PD, and DmDcr-2/R2D2/Loqs-PD complexes.

Minor points

1. Missing references in the text. For example, there are no citations for these two below statements.
"These cofactors bind with Dicer at a stoichiometric ratio and affect its cleavage activity, product length, and product delivery etc." "Recent studies have revealed the structure and function of DmDcr-2/Loqs-PD and DmDcr-2/R2D2 complexes separately."
2. Line 133 should be 19-bp dsRNA not 50-bp dsRNA
3. The cited figures are not in the right order. For example, the Extended Data Fig. 3d was mentioned before Extended Data Fig. 3a, b, c. And the Extended Data Fig. 3b before the Extended Data Fig. 3a.

Overall response

We are quite delighted that the reviewers thought the properties of Dicer oligomerization to be interesting and we thank the referees for their comments, which have substantially strengthened the manuscript. Referring to the super constructive suggestion raised by all the reviewers, we have added extensive experimental data and substantially revised the manuscript. We have made several modifications to the manuscript. 1) We have included investigations into the mechanisms of oligomer aggregation and depolymerization in vitro. 2) We have removed the in vitro FRAP experiments as we discovered that other components can also form similar patterns 3) and eliminated the inconclusive discussion on the potential connection between Dicer2 and RLRs, as although they share structural similarities, we could not establish a definitive link between the two. Below is our tentative response to the reviewers' comments (cited in italics).

Reviewer #1 (Remarks to the Author):

Manuscript by Deng and colleagues describes cryo-EM structures of Dm Dicer-2 with R2D2, Loqs-PD and two dsRNA helices (50- and 19-bp long). Understanding the roles of R2D2 and Loqs-PD in Dicer-2 function of dsRNA cleavage and product loading onto Ago2 has been the rationale for this work and has not been addressed by previous cryo-EM studies. Surprisingly, the authors report that DmDcr2 complexed with dsRNA and R2D2/Loqs-PD undergo di- or tri-merization, and FRAP experiments suggest high stability of the oligomers.

While the structures of DmDcr-2 complexes with R2D2, Loqs-PD (only the C-terminus of which is resolved) and dsRNA could provide interesting mechanistic insights into Dcr-2 function and the observation of oligomers is intriguing, the manuscript is disappointingly shallow as it lacks a deeper analysis of mechanistic implications into the roles of R2D2, Loqs-PD (and comparison with other Loqs- proteins and Dcr-1) and the role of oligomerization. A large body of work is neglected, e.g. the biochemical data discussing the possible roles of Dcr2 oligomerization vs. processivity on long dsRNA (for example, see <https://www.ncbi.nlm.nih.gov/pmc/articles/PMC3061311/> and <https://www.ncbi.nlm.nih.gov/pmc/articles/PMC3115569/> as well as subsequent manuscripts by these and other groups). As such, the manuscript needs to be substantially revamped to address the findings in the context of the existing literature. Additional detailed criticisms below must be addressed to improve the revised manuscript.

We are grateful to the reviewer for carefully reading our manuscript and making several necessary suggestions. References and others mentioned have been integrated into the new manuscript. The reviewer's detailed comments are addressed below.

1. Cryo-EM structures of Dcr-2 with R2D2 (<https://pubmed.ncbi.nlm.nih.gov/35768503/>) and with Loqs-PD (<https://www.nature.com/articles/s41586-022-04911-x>) were recently reported. Deng et al seem to argue that oligomerization of Dcr-2 requires both R2D2 and Loqs-PD. But Loqs-PD is bound at the periphery of Dcr-2 and is not engaged in oligomer formation. Why does the current study observe Dcr-2 oligomers as opposed to the Dcr-2*R2D2 structure? What are the functional implications of both studies (this paper and Yamaguchi et al) and the differences observed between the studies?

Response: We prepared cryo-EM samples without Loqs-PD according to your suggestion, but we found that in the absence of Loqs-PD, DmDcr-2/R2D2 and RNA remained mostly in the monomeric state and only a small fraction underwent oligomerization, so Loqs-PD is important for complex oligomerization. Due to the low proportion of oligomers, we were unable to determine their three-dimensional structures and consequently, unable to confirm whether the oligomers are identical with or without Loqs-PD.

We noted that in the absence of Loqs-PD, the helicase of the Dcr-2*R2D2 complex did not undergo the conformational change necessary for oligomerization, resulting in the inability of the complex to further aggregate. These findings strongly suggest the importance of Loqs-PD in inducing the required conformational change for oligomer formation.

Our revised manuscript includes relevant data and discussions to support these findings in **Extended Data Fig. 8**.

Response Fig. 1. **a**, 2D class-averages of DmDcr-2/R2D2/siRNA-duplex samples incubated with ATP and MgCl₂. **b**, Overview of the Helicase domain in the 1st monomer of the 19bp-dsRNA trimer state and **c**, the DmDcr-2/Loqs-PD/siRNA complex (PDB:7V6C). The dsRNA duplex alone cannot induce the conformational changes of the Helicase-DUF283 domains in DmDcr-2/R2D2 complex that are necessary for the oligomeric process. Therefore, Loqs-PD is crucial in initiating the oligomerization process.

2. Only the Hel2-binding region of Loqs-PD was observed in this and previous work. The recent Dcr-1*Loqs-PB structure finds all three domains of Loqs-PB interacting with the Dicer and RNA substrate

(<https://pubmed.ncbi.nlm.nih.gov/36182693/>). Discuss functional/structural differences between Loqs-PB and Loqs-PD that might lead to the lack of Loqs-PD interactions with dsRNA or Dcr-2.

Response: Loqs-PB and Loqs-PD have the same N-terminal 337 amino acids which contains two type A dsRBDs. In the recent study of Loqs-PB, these two dsRBDs binding the pre-miRNA substrate and promote the cleavage. It suggests that the function of two dsRBDs in Loqs-PD is also dsRNA-binding. However, previous single molecule study shows that Loqs-PD enhance the cleavage affinity by simply increasing the initial dsRNA-binding frequency of DmDcr-2, it does not change the processivity of DmDcr-2. It promotes the formation of initial dsRNA-binding state, and probably dissociated from dsRNA before that. Otherwise there the binding of dsRBDs could inhibit the DmDcr-2/Loqs-PD complex translocation along dsRNA, which is not observed in single molecule study. And the structural difference of the C-terminal region between Loqs-PB and Loqs-PD causes the distinct spatial location of the N-terminal dsRBD, which lead to the functional difference of them.

Response Fig. 2. Structural comparison of *DmDcr-2/Loqs-PD* apo state and *DmDcr-1/Loqs-PB/pre-miRNA* complex, aligned by the Hel2 sub-domain. The Loqs-PD is modeled based on the apo state of the *DmDcr-2/Loqs-PD* complex (PDB:7W0B) and the AlphaFold2-predicted structure of Loqs-PD.

3. Line 109: “Same as the previous structures” – which ones? Change to “As in previous studies (references), only the small segment ... is defined in our maps...”?

Response: Thanks for the suggestion. Thank you for your suggestion. We have followed your advice and rectified the statement that was causing confusion.

4. Line 159: “R2D2 and Loqs-PD bind to the Helicase domain of the 1st *DmDcr-2* at the same time...” – This must be a mistake, as the timing of their binding is

unknown from this work. The proteins likely bind sequentially rather than simultaneously.

Response: Sorry for the confusing description. We have rewritten this sentence in the revised manuscript, as below: “R2D2 and Loqs-PD bind to the Helicase domain of the 1st *DmDcr-2* together...”.

5. Line 171 “this region is missing in all the other solved structures”. Do you mean that the region was not modeled in the structures? Or that the sequence of the region is missing?

Response: Sorry for the confusing description. We have rewritten this sentence in the revised manuscript, as below: “...this region is not modeled in other solved structures of Dicer proteins”.

6. Lines 178-190. In this paragraph, the authors discuss the “translocation” and “active dicing” states. How are these states defined structurally? An introduction explaining the structural differences, and why they are interpreted as these functional states, would be helpful.

Response: Sorry for this incomplete description, we have added relevant content in the introduction section: “...*DmDcr-2* has been observed in several conformational states during the cleavage of dsRNA substrates, including the translocation state, active dicing state, and post-dicing state, among others...”

7. Line 185 “Two loop forms” – unclear.

Response: Sorry for the unclear description, we have rewritten this sentence in the revised manuscript, as below: “The newly built loops (aa 1551-1559 and aa 1594-1608) form a hydrophobic surface...”.

8. From FRAP experiments, it is not clear how the roles of R2D2, Loqs-PD and dsRNA were tested in oligomerization of Dcr-2. FRAP experiments with individual cofactors, and with their combinations must be performed to address this point.

Response: After following these suggestions, we conducted further research and discovered that the formation of large aggregates is not influenced by R2D2 and Loqs-PD, but by ATP and RNA. As a result, the previous conclusion regarding the mechanism of oligomer formation observed through fluorescence microscopy is inaccurate. We have removed this section from our revised manuscript.

9. In Discussion, the authors suddenly talk about the “C-shape” and “O-shape”

states. It would help the reader if the functional meaning and the structural differences are discussed.

Response: Sorry for this incomplete description, we have added relevant content in the revised manuscript (**Extended Data Fig. 8**).

10. Misleading jargon (like “Loqs-PD grabs the surrounding RNA”) should be avoided. Most structural mechanisms involve stochastic interactions and stochastic movements, rather than directed purpose-driven actions (such as “grabbing”). Furthermore, the authors do not demonstrate any evidence that Loqs-PD interacts with RNA. Improve the discussion by considering that interactions are stochastic; discussion should include the analysis of the data in this work and in previous genetic, biophysical, biochemical and structural studies.

Response: Thanks for the suggestion. We have corrected these misdescriptions in the revised manuscript.

Reviewer #2 (Remarks to the Author):

Drosophila Dicer2 produces siRNA to prevent viral infection and protect genome integrity. Two cofactors, R2D2 and Loqs-PD, bind to Dicer2 to aid processing the dsRNA substrate and loading the siRNA product to Ago2, respectively. The authors solved the structures of the Dicer2-Loqs-PD complex and revealed how the dsRNA substrate are recognized, translocated, and processed in the previous study. Others solved the structures of the Dicer2-R2D2 complex and explained the mechanism underlying the strand selection of the siRNA product. In this study, the authors determined the structures of Dicer2 in the presence of both Loqs-PD and R2D2. The major finding is that Loqs-PD induces R2D2-mediated oligomerization of Dicer2. High resolution view of the specific interactions of two cofactors with Dicer2 suggests evolutionary designed functions of two cofactors, which has been unknown so far. The oligomerization of Dicer proteins by cofactors has never been shown and is an interesting feature of Drosophila Dicer2. However, it is unknown what is the advantage of the oligomerization or whether the oligomerization state exists in vivo. Therefore, the physiological importance of the oligomerization and thus its detailed mechanism presented in the manuscript is unclear. While the oligomeric structure is interesting and potentially important, more extensive structural analyses including detailed comparison with previous structures of Dicer2 are recommended to make hypotheses and conclusions reasonable and understandable. In addition, there are too many typos and grammar errors. A professional English editing service is highly recommended.

We would like to thank the reviewer for the very careful reading of our manuscript and the numerous and constructive comments, and we tried to address all the proposals in the revised manuscript.

The major points

1. The introduction is not sufficiently provided to fully understand the presented structures and discussions. For example, the "translocation state" has never been explained in this manuscript. Without previous knowledge, one could easily think it is related to the cellular localization of Dicer2, not the translocation of RNA in the protein complex. The previous work by the authors and others need to be described in details for broad readership.

Response: We are very grateful for these suggestions. We have added a description of the dicing process in the new introduction and improved some of the previous work related to the content of the article.

2. The authors need to provide at least one evidence that shows physiological importance of the oligomerization. For examples, one could observe the oligomeric state of Dicer2 in cells or tissues or investigate the effect of a mutation in R2D2 that prevent the Dicer oligomerization on the processing

activity and selectivity in cells.

Response: Due to our technical limitations, we are currently unable to provide the requested evidence. The fluorescence experiments have revealed that these three protein components are capable of co-localizing *in vivo*, but we were unable to determine if oligomers were formed. The negative staining data suggests that these three components can easily form oligomers in the presence of dsRNA and ATP. However, at low concentrations, the proportion of fiber formation is low (as shown in **Figure 4b**) and detecting oligomers under fluorescence microscopy proves to be challenging. We also conducted a mutation on the "bridge helix" of R2D2, however, this cannot prevent the formation of oligomers. Nevertheless, the mutation may alter oligomer rigidity.

3. Without RNA, what was the structure of the protein complex like? Was it the 1:1:1 complex of Dicer2, R2D2, and Loqs-PD? If this is true, after the incubation with RNA, the majority of Loqs-PD proteins should have been dissociated. Then, how was Loqs-PD dissociated? Simply by the difference of the binding affinity? The authors need to include discussion about how the oligomeric state is formed during the cryo-EM sample preparation.

Response: When oligomerization does not occur, the binding sites of Loqs-PD and R2D2 on the same Dcr-2 molecule do not experience spatial conflicts. Our SEC and SDS-page results suggest that the three proteins assemble in a 1:1:1 ratio in the absence of RNA. We proposed a hypothesis in the discussion, based on the properties of oligomers, that releasing Loqs-PD could facilitate the transmission of more cleavage products to R2D2. However, we are unable to provide a detailed explanation for the oligomerization process, including the mechanism of Loqs-PD release.

4. The binding region of 50-bp-RNA in Dicer-2 is overlapped with the oligomeric interaction site. Then, how could higher order oligomers be formed as shown in Fig 4a. Is this also related to the smaller size of the oligomer in the presence of 50-bp-RNA compared with that in the presence of 19-bp-RNA?

Response: We agree with the view that 50-bp-RNA is theoretically not able to form higher order oligomers due to spatial overlap, only short dsRNAs can. According to the analysis of two sets of cryo-EM data, we obtained dimers only in the 50-bp-RNA, and there is a subtle difference in the mode of polymerization between dimers and trimers.

5. The authors mentioned the the oligomerization may increase the processing efficiency by increasing local protein concentration. However, the oligomerization appeared to stop the processing cycle and hold Dicer at the

translocation conformation. Then how can the oligomerization increase the efficiency in the oligomeric state? The discussion seems contradictory.

Response: We are very sorry for these uncritical discussions, and after an in-depth structural analysis, we have removed the statement.

6. There are many typos and grammar errors. A professional English editing service is highly recommended.

Response: We are very grateful for this suggestion. We have revised some typos and grammar errors accordingly and polished the writing throughout the manuscript.

The minor points

Line 31. It is difficult to understand the connection between large oligomers and a common evolutionary origin. Does this mean that many proteins in RNA-induced innate immunity oligomerize? Then this needs to be more properly described.

Response: After in-depth literature research work, we have removed this part.

Line 58 and 60. The functions of TRBP, R2D2, and Loqs proteins need to be explained before these sentences. No description on their functions is included in the introduction at all.

Response: Thanks for your suggestion and we have added some notes about the function of these proteins in the new revised manuscript.

Line 64. The features of the substrate of Dicer2 and how they are processed need to be described before this section.

Response: Thanks for your suggestion, we have added this section to introductions.

Line 71. The author described "it was reported that Loqs-PD and R2D2 act sequentially in the siRNA pathway". How solid would be this data? What would be still unclear? Since the presented work showed they can act simultaneously, the authors could raise some questions from this previous data.

Response: Although we observed co-localization of these three proteins *in vivo*, we cannot conclusively demonstrate that they consistently bind together. It is possible that Loqs-PD and R2D2 are regulated during their active state, leading

to sequential binding with *DmDcr-2*.

Line 78. References need to be cited.

Response: The error has been corrected.

Line 95. This sentence can be mistaken for that protein:ATP is 1:1.

Response: Sorry for the confusing description. We have rewritten this sentence in the revised manuscript, as below: "The protein complexes and RNAs were then incubated at equal molar ratios in buffer containing ATP and Magnesium ions".

Line 106. Loqs-PD and R2D2 have large regions missing in the cryo-EM map. These need to be more clearly described in the text and shown in Fig 1f (e.g. by representing the missing part as gray color)

Response: Thanks for the suggestion. We have modified fig.1f according to your suggestion.

Line 129. How the different role would be possible needs to be explained here based on the structural comparisons.

Response: We have added "...that it can change the aggregation direction of *DmDcr-2*" following the sentence.

Line 131. This paragraph is confusing, since there was no description that the trimeric structure was solved for both 19bp- and 50bp-dsRNA. Line 100 misleads the readers to understanding that only one structure for each dsRNA substrate was solved.

Response: Sorry for the confusing description. We have rewritten this sentence in the revised manuscript, as below: "...we obtained two structures from the 50bp-complex dataset: one in the dimer state with two *DmDcr-2* monomers (3.72 Å) and one in the trimer state with three *DmDcr-2* monomers (3.74 Å), as well as one structure from the 19bp-complex with three *DmDcr-2* monomers in the trimer state (3.70 Å)".

Line 139. Trimer need to be specified to the trimer with 50-bp-RNA.

Response: Since the two Trimer structures from different dataset are almost identical, we do not specify them again.

Line 141. The sentence is difficult to understand. Please rephrase it.

Response: Sorry for the confusing description. We have rewritten this sentence as below: "This difference is also supported by the relative distances and angles between adjacent *DmDcr-2* monomers, which are all very similar in the trimeric state".

Line 159. Do they really bind at the same "time"? "together" may be better.

Response: Thanks for the suggestion. We replaced them with "together".

Line 165. What restraint did the authors mean?

Response: Sorry for the confusing description. We have interpreted the sentence: "...as R2D2 bound to the first *DmDcr-2* can inhibit the second *DmDcr-2* from binding to Loqs-PD".

Line 168. Please give an information that RNase IIIs have two long insertions before this

Response: Sorry for the incomplete description. We have added relevant content and references: "...An insertion region of unknown function is often present within the RNaseIII domain such as Dorsha or Dicer. Sequence alignment results show that both the sequences of insertion region of RNase IIIa (RIIIDai) and RNase IIIb (RIIIDbi) are not conserved among Dicer proteins..."

Line 180. "translocation state" needs to be defined somewhere before this sentence.

Response: Thanks for the suggestion. The revised manuscript now includes related content.

Line 191. This hypothesis seems easy to test. The authors already have all materials.

Response: The revised manuscript contains the results of the experiment (**Fig.3 g-h**).

Line 200. How is the complex labeled? by Alexa488 antibody or NHS? No information is found even in Methods.

Response: Since this part of the experiment needs further confirmation, we have removed this part.

Line 223. "Therefore, Loqs-PD may promote the conformational change of Helicase domain." Could this be explained by the previous structures of the Dicer2-Loqs-PD complex?

Response: The conformation of Helicase in the oligomer is the same as that of the bound RNA in the previous *DmDcr-2/Loqs-PD* structure. It can also be explained upon comparison with the *DmDcr-2/R2D2* structure.

Line 227. Is it confirmed that Loqs-PD initializes the oligomerization process? It seems that dsRNA, not Loqs-PD, initializes the oligomerization, based on the sample preparation protocol.

Response: Comparing the structure of *Dcr-2/R2D2/siRNA-duplex* (PDB-7V6C) with that of *Dcr-2/Loqs-PD/dsRNA* (PDB-7W0A), Helicase did not undergo the conformational change necessary for oligomerization when only dsRNA was available, so Loqs-PD may initiate the oligomerization process by causing the conformational change of Helicase. The two-dimensional classification by cryo-EM showed that without Loqs-PD, most of the particles remains as monomers in **Extended Data Fig.8e**.

Response Fig. 3. **a**, Overview of the Helicase domain in the 1st monomer of the 19bp-dsRNA trimer state and **b**, the *DmDcr-2/Loqs-PD/siRNA* complex (PDB:7V6C). The dsRNA duplex alone cannot induce the conformational changes of the Helicase-DUF283 domains in *DmDcr-2/R2D2* complex that are necessary for the oligomeric process. Therefore, Loqs-PD is crucial in initiating the oligomerization process.

Line 240-243. The discussion is difficult to understand. More proper phrasing and background are required.

Response: As there is no concrete evidence to support their relationship, the discussion regarding the RLRs family has been removed.

Line 323. The specific information of resins and columns need to be provided. Especially, the size of the protein complex cannot be estimated due to no information of SEC column.

Response: The methods of revised manuscript now include related information.

Fig 3 legend includes an incomplete sentence.

Response: The error has been corrected.

Extended Data Fig. 5 legend. The alignment contains three green boxes. Which is the insertion mentioned in Line 168?

Response: Thanks for the suggestion. We have made a detailed explanation.

Extended Data Fig. 1a. Reference molecular weights need to be indicated to estimate the sizes of the protein peaks.

Response: Thanks for the suggestion. Now Extended Data Fig. 1a includes the related information.

Extended Data Fig. 6 is difficult to see how the dicer conformation and the R2D2 location change. The figure sizes need to be increased. The structure of 7V6C need to be colored by domains and proteins.

Response: Thanks for the suggestion. We have made improvements based on your suggestions in **Extended Data Fig. 8** now.

Reviewer #3 (Remarks to the Author):

In this manuscript (MS), Deng and colleagues solved the structures of DmDcr-2/R2D2/Loqs-PD/dsRNA using the Cry-EM approach. They found a dimer of DmDcr-2/R2D2 and one Loqs-PD on 50-bp dsRNA and a trimer of DmDcr-2/R2D2 and one Loqs-PD on 19-bp dsRNA. Although the cryo-EM structure of DmDcr-2/R2D2 and DmDcr-2/Loqs-PD were already solved, the structure of the ternary complex of DmDcr-2/R2D2/Loqs-PD in dsRNA solved in this study is a great addition to our understanding of complex assembly involving DmDcr-2 and its cofactors. The structures are at high resolution, and the description and interpretation of the structures are accurate and comprehensive. However, I have several major comments below.

We are grateful for the supportive comments and would like to address concerns raised by the referee in the below point-by-point response.

Major points

1. The study did not address functions of the oligomer states of the DmDcr-2/R2D2/Loqs-PD in the DmDcr-2 processing mechanism and other cellular functions of DmDcr-2. Why does the enzyme complex need to be dimerized on the substrate, 50-bp dsRNA? Why does the enzyme complex need to be trimerized on the cleaved product, 19-bp dsRNA?

Response: *DmDcr-2* used here had no dicing active, and long dsRNA can be cleavage by WT *DmDcr-2*, so it is highly possible that the 50-bp-RNA formed dimer is not present *in vivo*. Our hypothesis is that the oligomer can safeguard the siRNA duplex from degradation and enable its loading onto Ago2.

2. What are the cells' oligomer states of DmDcr-2/R2D2/Loqs-PD?

Response: Currently, our technology has not been successful in detecting the presence of these oligomers in the S2 cell line. As a result, we do not have direct data on the status of their presence *in vivo*.

3. In the Extended Data Fig. 1, R2D2, smaller than Loqs PD, had a higher density in the gel than Loqs-PD. This indicated that the assembled DmDcr-2/R2D2/Loqs-PD already contained more DmDcr-2/R2D2 than Loqs-PD. What is the oligomer state of DmDcr-2/R2D2/Loqs-PD without RNAs?

Response: We co-expressed and purified *DmDcr-2/R2D2* and assembled the complex with Loqs-PD. Due to this process, the ratio of Loqs-PD in the final complex may be lower than R2D2. We attempted to obtain the three-dimensional structure of *DmDcr-2/R2D2/Loqs-PD* without RNAs using cryo-EM analysis but were unsuccessful. Nevertheless, based on our observation of the 2D averages, we note that while there were various conformations present, the

majority of them appeared to be in the monomeric state.

Response Fig. 4. 2D averages of DmDcr-2/R2D2/Loqs-PD without RNAs.

4. In figure 4, they found that *DmDcr-2/R2D2/Loqs-PD/dsRNA* complexes can form huge irregular oligomers in the presence of ATP and Mg⁺⁺ *in vitro*. 1) Are the dimeric and trimeric states of the complexes shown in Fig. 1 just a small portion of the various oligomer state? 2) What are the oligomer states of *DmDcr-2/R2D2/Loqs-PD* complexes without RNA in the presence of ATP and Mg⁺⁺ *in vitro*? Whether RNAs induced the complex oligomerization?

Response: 1) The state of aggregation is generally concentration-dependent, and the concentration of the sample prepared for EM is much lower than that for FRAP experiments. Therefore, it is difficult to determine the proportion of dimer or trimer in it. 2) Negative-staining EM micrograph analysis reveals that *DmDcr-2/R2D2/Loqs-PD* complexes do not undergo oligomerization in the absence of RNA (**Extended Data Fig. 6**). Thus, RNA is necessary for oligomerization and only in the presence of RNA can the Helicase domain undergo the necessary conformational changes for oligomer formation.

Response Fig. 5. One of negative-staining EM micrograph and 2D averages of *DmDcr-2/R2D2/Loqs-PD* complexes with 5mM ATP and 5mM MgCl₂.

5. They claimed that "*R2D2* and *Loqs-PD* can bind to different regions of *DmDcr-2* without spatial clash". How could they always find only one *Loqs-PD* in either dimeric or trimeric *DmDcr-2* containing 2 (or 3) of *R2D2* + 2 (or 3) of

DmDcr-2?

Response: Sorry for the not clear enough descriptions, For the DmDcr-2 monomer, the binding sites of Loqs-PD and R2D2 does not conflict. Only after the oligomer formation, R2D2 in the first monomer (1st R2D2 in Fig. 1a) conflicts with Loqs-PD at the binding site of the 2nd Dcr-2, thus inhibiting the binding of Loqs-PD to the 2nd Dcr-2.

6. Since the authors obtained DmDcr-2/R2D2, DmDcr-2/Loqs-PD, and DmDcr-2/R2D2/Loqs-PD complexes and they claimed that the structures they received were not DmDcr-2 in "active dicing" state. And they thought that R2D2 traps DmDcr-2 in the "translocation state." The authors should compare the dicing activity of DmDcr-2/R2D2, DmDcr-2/Loqs-PD, and DmDcr-2/R2D2/Loqs-PD complexes.

Response: Thanks for the suggestion. The dicing activity assay has been included in our revised manuscript. These complexes were ranked according to cleavage activity as: Dcr-2/Loqs-PD > Dcr-2/R2D2/Loqs-PD > Dcr-2 > Dcr-2/R2D2.

Response Fig. 5. a, Cleavage assays of DmDcr-2, DmDcr-2/Loqs-PD, DmDcr-2/R2D2, DmDcr-2/R2D2/Loqs-PD with 106-bp dsRNA (0.3 μ M) with ATP, in the cleavage assay buffer (100mM Tris-8.0, 100mM NaCl, 1mM DTT, 5mM MgCl₂, 25 °C for 30min). Products were resolved on a 16% polyacrylamide denaturing gel. **b**, The plot of cleavage efficiency (n=3 each; means \pm SD).

Minor points

1. Missing references in the text. For example, there are no citations for these two below statements. "These cofactors bind with Dicer at a stoichiometric ratio and affect its cleavage activity, product length, and product delivery etc." "Recent studies have revealed the structure and function of DmDcr-2/Loqs-PD and DmDcr-2/R2D2 complexes separately."

Response: Thanks for the suggestion. We have revised and updated

manuscript accordingly.

2. Line 133 should be 19-bp dsRNA not 50-bp dsRNA

Response: We have corrected these errors in the revised manuscript.

3. The cited figures are not in the right order. For example, the Extended Data Fig. 3d was mentioned before Extended Data Fig. 3a, b, c. And the Extended Data Fig. 3b before the Extended Data Fig. 3a.

Response: Thanks for the suggestion. We have rearranged the figure panels in the revised manuscript.

REVIEWER COMMENTS

Reviewer #1 (Remarks to the Author):

This is a revised manuscript by Deng et al, describing cryo-EM structures of *Drosophila* Dicer-2 (catalytically inactive mutant) in complex with Loqs-PD and R2D2 with either 19-bp or 50-bp dsRNA. The authors report a curious finding that Dicer-2 is capable of binding Loqs-PD and R2D2 simultaneously (at non-overlapping sites) and such complexes form head-to-tail oligomers, in which dsRNA is positioned to thread through two or three copies of Dicer-2. This is a noteworthy observation, but it remains unclear whether the oligomers represent a functionally relevant state(s). To complicate the matter further, the authors' biochemical data and previous studies are not consistent with structural implications that the authors draw.

Oligomerization of Dcr-2*R2D2*Loqs-PD may correspond to several functions: (1) activation of a pathway, such as: (a) enhancing dsRNA cleavage (relative to monomers); (b) enhancing siRNA loading onto Ago-2, via R2D2. (2) Alternatively, the oligomers may represent inactivated (e.g. stress-induced) states. (3) The third possibility is that the oligomers can assemble stochastically without substantially affecting any pathway. These possibilities are not clearly discussed in the manuscript. The authors hint that the oligomer may represent a state that is less active than monomeric Dcr-2: the oligomers bind R2D2, which slows down the dicing reaction (Fig. 3h). On the other hand, oligomerization is enhanced by ATP and is inhibited by ADP. ATP binding and hydrolysis catalyze long dsRNA dicing, so does the oligomerization enhancement by ATP imply that oligomerization enhances the rate of dicing? Either way, kinetic assays could provide insight into the role of oligomerization in dicing. Did the authors fit their kinetic data into the Hill equation, which could show if cooperativity plays a role in the catalysis? Along these lines, it is interesting to ask whether oligomers affect single-turnover vs multiple-turnover reactions, i.e. whether product release is affected by oligomerization. Finally, the ATP hydrolysis state/function in oligomers remain unaddressed – see below.

Addressing these conceptual questions would result in a much more insightful manuscript with a clear (rewritten) Discussion/conclusions.

Other criticisms/comments are listed below:

1. The authors formed the complex with ATP. What does the cryo-EM density in the ATPase center correspond to: ATP, ADP-Pi or ADP? What are the mechanistic implications of this finding?
2. On line 99, the authors state that “synergistic interactions ... with Loqs-PD and R2D2 are still unclear”. Which process do the authors refer to, in which Loqs-PD and R2D2 can act synergistically? How does the observation of oligomers in this work address synergy?
3. The authors describe the intermolecular contacts that may stabilize the oligomer (ca ~line 132). For example, the C-terminus of R2D2 interacts with two adjacent Dcr-2 copies, “suggesting its vital role...”. How conserved is the C-terminus? How conserved are the contact sites on both Dcr-2 copies? Do mutations of the R2D2 or Dcr-2 residues that stabilize the interactions disrupt oligomerization?
4. Line 217: “No other Dicer protein oligomerization has been reported perhaps due to the difference in this region”. Do the authors mean the sequence difference? If this region is not conserved and oligomerization is not observed for Dcr-2 homologs (in insects/invertebrates?), does DmDcr-2 oligomerization represent an idiosyncratic phenomenon, which is unlikely to have a conserved function?

Alternatively, is there a Dm- or insect-specific function in oligomerization?

5. Methods must be revised (extended) to ensure that these experiments can be reproduced. There are several areas that are not well documented. For example, (1) purification of Loqs-PD is not described. Purification of Dcr2-R2D2 is barely described – which “anion exchange” method was used? (2) buffer composition for individual proteins storage is not reported (3) What were the concentrations of complex components? (this sentence does not make sense “were mixed at a molar concentration of 1:1.5” because no concentration is specified. (4) In cryo-EM grid preparation, what buffer was used for dilution (“For each sample, an aliquot of 4 μ l of purified samples was diluted to 1 μ M”)? These are just a few examples – many other places have to be expanded or clarified to enable the reproduction of this work.

6. Overall, further English proofreading is needed, as several stylistic errors, unclear statements and misspellings have to be corrected.

Reviewer #2 (Remarks to the Author):

The authors addressed most of my concerns and the manuscript has been largely improved. However, some major issues are still remained to be addressed.

1. Regarding the major point #2, the authors could not provide any evidence that shows physiological importance of the oligomerization. While the difficulty to obtain the in vivo data is understandable, since this is an important issue, I like to suggest an alternative. If this oligomerization is functionally important, the residues involved in the oligomerization should be evolutionarily conserved at least in insects. An extensive sequence alignment of Dicer 2 homologs may find substantial sequence conservation in the residues, and this would strengthen the manuscript.

2. Several issues regarding the new data in Figure 3b:

1) Too little information is provided for the processing experiment in the text. Please explain the aim of the experiment, what are the substrates, and what products were expected to be produced and have actually been produced.

2) Dicers generally cut a specific cleavage site producing specific sizes of products. Why is the size range of the cleaved products so wide? How are the authors sure that they are Dicing products, not by RNase impurities?

3) No method for this experiment is provided at all.

Minor comments

1. In Abstract, the sentence "The formation and depolymerization of oligomers are associated with ATP"

is unclear. Which of ATP energy, binding, or hydrolysis needs to be specified.

2. Line 63, bind "to"

3. Line 70, bind "to"

4. Line 74, are dsRBDs completely independent to RNA sequences? those in bacterial RNaseIII and Drosha have substantial sequence preferences.

5. Line 225, a typo.

6. Line 279, Difficult to understand the main point of this paragraph.

7. Line 297, is  are

Reviewer #3 (Remarks to the Author):

Thank you for your previous responses, and I appreciate the information you have provided.

In response to my second question, you mentioned that your current technology has not been successful in detecting the presence of these oligomers in the S2 cell line, and as a result, you do not have direct data on their presence in vivo. I encourage you to explore alternative methods or techniques to address this question, as it would significantly strengthen the conclusions of your study. Investigating this aspect will help confirm the existence of the oligomerization state of the enzyme complex in cells and provide insight into its cellular functions. It is possible that the well-demonstrated oligomerization state in the current study is a result of non-cellular complex reconstitution.

Overall response

We highly appreciate all the reviewers' comments and tremendous efforts in revising our manuscript. Main modifications are marked in red text in the revised manuscript. Concerns and suggestions are carefully addressed in the following text.

Reviewer #1 (Remarks to the Author):

This is a revised manuscript by Deng et al, describing cryo-EM structures of Drosophila Dicer-2 (catalytically inactive mutant) in complex with Loqs-PD and R2D2 with either 19-bp or 50-bp dsRNA. The authors report a curious finding that Dicer-2 is capable of binding Loqs-PD and R2D2 simultaneously (at non-overlapping sites) and such complexes form head-to-tail oligomers, in which dsRNA is positioned to thread through two or three copies of Dicer-2. This is a noteworthy observation, but it remains unclear whether the oligomers represent a functionally relevant state(s). To complicate the matter further, the authors' biochemical data and previous studies are not consistent with structural implications that the authors draw.

*Oligomerization of Dcr-2*R2D2*Loqs-PD may correspond to several functions: (1) activation of a pathway, such as: (a) enhancing dsRNA cleavage (relative to monomers); (b) enhancing siRNA loading onto Ago-2, via R2D2. (2) Alternatively, the oligomers may represent inactivated (e.g. stress-induced) states. (3) The third possibility is that the oligomers can assemble stochastically without substantially affecting any pathway. These possibilities are not clearly discussed in the manuscript. The authors hint that the oligomer may represent a state that is less active than monomeric Dcr-2: the oligomers bind R2D2, which slows down the dicing reaction (Fig. 3h). On the other hand, oligomerization is enhanced by ATP and is inhibited by ADP. ATP binding and hydrolysis catalyze long dsRNA dicing, so does the oligomerization enhancement by ATP imply that oligomerization enhances the rate of dicing? Either way, kinetic assays could provide insight into the role of oligomerization in dicing. Did the authors fit their kinetic data into the Hill equation, which could show if cooperativity plays role in the catalysis? Along these lines, it is interesting to ask whether oligomers affect single-turnover vs multiple-turnover reactions, i.e. whether product release is affected by oligomerization. Finally, the ATP hydrolysis state/function in oligomers remain unaddressed – see below. Addressing these conceptual questions would result in a much more insightful manuscript with a clear (rewritten) Discussion/conclusions.*

Response: We greatly appreciate the suggestions provided in the review. Oligomerization can increase local concentration and enhance activity or activate some signaling pathways. However, Dcr-2 oligomerization does not enhance cleavage activity, and there is currently a lack of report on Dicer

oligomerization. Oligomers seem to have a closer relationship with cleavage products and can protect them from degradation, which leads us to believe that the oligomers may temporarily protect cleavage products when Ago2 is absent. The kinetic assays are of significant help in understanding the function of the Dcr-2/R2D2/Loqs-PD complex. We apologize that, due to constraints in our current resources, we are unable to complete them within a short timeframe. For example, **ordering isotopically labeled reagents requires more than four months in our institute.** We will conduct relevant experiments based on the reviewer's suggestions and hope to include this part in future research.

Other criticisms/comments are listed below:

1. The authors formed the complex with ATP. What does the cryo-EM density in the ATPase center correspond to: ATP, ADP-Pi or ADP? What are the mechanistic implications of this finding?

Response: **Additional density can be observed at the ATPase center of the Helicase domains,** but it cannot be determined whether it is ATP or ADP based on the density alone (**Response Fig. 1**). As the center is in an active state, the density may be either ADP or ATP undergoing hydrolysis. We propose that after dsRNA binds to the Helicase domain, steric hindrance may occur, which affects oligomerization, so ATP hydrolysis is required to adjust the binding position of dsRNA.

Response Fig. 1. a, The ATP binding sites of the trimeric structure are highlighted with dashed boxes. **b,** We placed an ADP and a magnesium ion at the ATP/ADP binding pocket based on the homologous structure (PDB-5E3H). However, it was not possible to determine whether it was ATP or ADP based on the density. Since it was in the active state, it is likely to be a mixture of ATP and ADP.

2. On line 99, the authors state that “synergistic interactions ... with Loqs-PD and R2D2 are still unclear”. Which process do the authors refer to, in which Loqs-PD and R2D2 can act synergistically? How does the observation of oligomers in this work address synergy?

Response: We found that the absence of either Loqs-PD (**Extended Data Fig. 10e**) or R2D2 (**Fig. 4a-b**) affects the formation of oligomers, indicating that Loqs-PD and R2D2 should act synergistically in this process. However, we do not have a detailed understanding of how they interact and function together during the oligomerization process.

3. *The authors describe the intermolecular contacts that may stabilize the oligomer (ca ~line 132). For example, the C-terminus of R2D2 interacts with two adjacent Dcr-2 copies, “suggesting its vital role...”. How conserved is the C-terminus? How conserved are the contact sites on both Dcr-2 copies? Do mutations of the R2D2 or Dcr-2 residues that stabilize the interactions disrupt oligomerization?*

Response: The sequence alignment results of diverse insects suggest higher conservation of the C-terminus of R2D2 only in *Drosophila* (**Extended Data Fig. 9c**). Due to resolution limitations, we could not identify the specific amino acids involved in the interaction between the "bridge helix" of R2D2 and the second Dcr-2. However, when we replaced R2D2's "bridge helix" with 8*GS, there is still a small proportion of oligomers, but the structure of the oligomers is not stable (**Extended Data Fig. 7a-d**).

4. *Line 217: “No other Dicer protein oligomerization has been reported perhaps due to the difference in this region”. Do the authors mean the sequence difference? If this region is not conserved and oligomerization is not observed for Dcr-2 homologs (in insects/invertebrates?), does DmDcr-2 oligomerization represent an idiosyncratic phenomenon, which is unlikely to have a conserved function? Alternatively, is there a Dm- or insect-specific function in oligomerization?*

Response: Sorry for the unclear description, we have deleted this sentence in the revised manuscript.

Through sequence alignment analysis, we observed that the RIIIDai of Dcr-2 and R2D2, which are located at the oligomerization interface, are predominantly conserved only in the Drosophilidae family (**Extended Data Fig. 9**). This suggests that *Drosophila* Dcr-2 is the only one that undergoes oligomerization, making it an idiosyncratic phenomenon specific to *Drosophila*.

Currently, *DmDcr-2* is the only well-studied protein among arthropods, and it is challenging to determine if oligomerization occurs in other insect Dcr-2 proteins due to the non-conservation of amino acid sequences.

5. *Methods must be revised (extended) to ensure that these experiments can be reproduced. There are several areas that are not well documented. For example, (1) purification of Loqs-PD is not described. Purification of Dcr2-R2D2*

is barely described – which “anion exchange” method was used? (2) buffer composition for individual proteins storage is not reported (3) What were the concentrations of complex components? (this sentence does not make sense “were mixed at a molar concentration of 1:1.5” because no concentration is specified. (4) In cryo-EM grid preparation, what buffer was used for dilution (“For each sample, an aliquot of 4 µl of purified samples was diluted to 1µM”)? These are just a few examples – many other places have to be expanded or clarified to enable the reproduction of this work.

Response: We thank the reviewer for bringing these issues to our attention and helping us improve the quality of our manuscript. We have corrected these points accordingly in the revised manuscript.

6. Overall, further English proofreading is needed, as several stylistic errors, unclear statements and misspellings have to be corrected.

Response: We thank the reviewer for these suggestions. We have thoroughly reviewed and revised the manuscript based on all the reviewers' comments to address the stylistic errors, unclear statements, and misspellings.

Reviewer #2 (Remarks to the Author):

The authors addressed most of my concerns and the manuscript has been largely improved. However, some major issues are still remained to be addressed.

1. Regarding the major point #2, the authors could not provide any evidence that shows physiological importance of the oligomerization. While the difficulty to obtain the in vivo data is understandable, since this is an important issue, I like to suggest an alternative. If this oligomerization is functionally important, the residues involved in the oligomerization should be evolutionarily conserved at least in insects. An extensive sequence alignment of Dicer 2 homologs may find substantial sequence conservation in the residues, and this would strengthen the manuscript.

Response: We appreciate the suggestions from the reviewer and have incorporated sequence alignment results in the revised manuscript. The results revealed that the RIIIDai of Dcr-2 and R2D2, which are located at the oligomerization interface, are conserved only in the Drosophilidae family, suggesting that oligomerization is unique to *Drosophila* Dcr-2 (**Extended Data Fig. 9**). While this is a fascinating discovery, the implications of these species-

specific differences require further investigation.

2. Several issues regarding the new data in Figure 3b:

- 1) Too little information is provided for the processing experiment in the text. Please explain the aim of the experiment, what are the substrates, and what products were expected to be produced and have actually been produced.*
- 2) Dicers generally cut a specific cleavage site producing specific sizes of products. Why is the size range of the cleaved products so wide? How are the authors sure that they are Dicing products, not by RNase impurities?*
- 3) No method for this experiment is provided at all.*

Response:

- 1) Both in the apo-state and translocation state, R2D2 interacts with Dcr-2 at multiple binding sites, which may affect the conformational change of Dcr-2 and thus inhibit its cleavage activity. Our hypothesis has been confirmed by vitro cleavage experiments, and the substrate sequence has been added to the new **Extended Data Fig. 6**.
- 2) The dsRNA used in our study was a non-labeled BLT-terminus dsRNA. Therefore, to enhance imaging, we loaded more RNAs in each lane and stained the gels with GelRed, which might have resulted in wider bands. We verified that all materials used in the experiments were nuclease-free. Moreover, unlike Dcr-2, RNase impurities usually do not have fixed bands.
- 3) Related methods are attached in the revised manuscript.

Minor comments

- 1. In Abstract, the sentence "The formation and depolymerization of oligomers are associated with ATP" is unclear. Which of ATP energy, binding, or hydrolysis needs to be specified.*
- 2. Line 63, bind "to"*
- 3. Line 70, bind "to"*

Response: We have addressed the above points in the revised manuscript.

4. Line 74, are dsRBDs completely independent to RNA sequences? those in bacterial RNaseIII and Drosha have substantial sequence preferences.

Response: While some dsRBDs exhibit sequence preferences, recent studies on human Dicer's dsRBD have shown differences in binding affinities for different types of dsRNA (<https://doi.org/10.1038/s41586-023-05722-4>). Nevertheless, these dsRBDs are capable of binding to other dsRNAs as well.

- 5. Line 225, a typo.*
- 6. Line 279, Difficult to understand the main point of this paragraph.*
- 7. Line 297, is  are*

Response: We have addressed the above points in the revised manuscript.

Reviewer #3 (Remarks to the Author):

Thank you for your previous responses, and I appreciate the information you have provided.

In response to my second question, you mentioned that your current technology has not been successful in detecting the presence of these oligomers in the S2 cell line, and as a result, you do not have direct data on their presence in vivo. I encourage you to explore alternative methods or techniques to address this question, as it would significantly strengthen the conclusions of your study. Investigating this aspect will help confirm the existence of the oligomerization state of the enzyme complex in cells and provide insight into its cellular functions. It is possible that the well-demonstrated oligomerization state in the current study is a result of non-cellular complex reconstitution.

Response: It is very significant to demonstrate that Dicer can oligomerize in vivo. We will conduct epigenetic experiments to verify the effect of oligomers on fruit fly development.

REVIEWERS' COMMENTS

Reviewer #1 (Remarks to the Author):

In the revised manuscript, most of my criticisms have been addressed. The manuscript is more balanced now that the discussion of the functional importance of the oligomers is toned down and the conservation of contact regions in insects is discussed. I do not have major criticisms. Yet the manuscript needs more English proofreading, as there are unfinished sentences and dependent clauses presented as sentences.

Reviewer #2 (Remarks to the Author):

The authors have addressed all of my concerns with the original manuscript. Although the authors did not clearly mention about the expected products in the RNA cleavage assay, I can clearly see ~19-nucleotide RNA products in the new gel data. I have no more comments.

We greatly appreciate the suggestions, as they have significantly enhanced the quality of the manuscript. Below is our tentative response to the reviewers' comments (cited in italics).

Reviewer #1 (Remarks to the Author):

In the revised manuscript, most of my criticisms have been addressed. The manuscript is more balanced now that the discussion of the functional importance of the oligomers is toned down and the conservation of contact regions in insects is discussed. I do not have major criticisms. Yet the manuscript needs more English proofreading, as there are unfinished sentences and dependent clauses presented as sentences.

Response: Thank you very much for your suggestions. In this version, with the assistance of ChatGPT and our colleagues, we have made every effort to improve the English proofreading as much as possible.

Reviewer #2 (Remarks to the Author):

The authors have addressed all of my concerns with the original manuscript. Although the authors did not clearly mention about the expected products in the RNA cleavage assay, I can clearly see ~19-nucleotide RNA products in the new gel data. I have no more comments.

Response: We apologize for the unclear labeling. The distortion of the gel image edges has made it difficult to establish a clear correspondence with the marker. However, based on the curvature trend of the markers, it can be inferred that the majority of the cleavage products have a length of approximately 21 nt.